# SELF-TUNING SELF-SUPERVISED ANOMALY DETECTION

## ABSTRACT

Self-supervised learning (SSL) has emerged as a promising paradigm that presents supervisory signals to real-world problems, bypassing the extensive cost of manual labeling. Consequently, self-supervised anomaly detection (SSAD) has seen a recent surge of interest, since SSL is especially attractive for unsupervised tasks. However, recent works have reported that the choice of a data augmentation function has significant impact on the accuracy of SSAD, posing *augmentation search* as an essential but nontrivial problem with the lack of labeled validation data. In this paper, we introduce ST-SSAD (Self-Tuning Self-Supervised Anomaly Detection), the *first systematic approach for rigorous augmentation tuning on SSAD*. To this end, our work presents two key contributions. The first is a new unsupervised validation loss that quantifies the alignment between the augmented training data and the (unlabeled) test data. Second, we present new differentiable augmentation functions, allowing data augmentation hyperparameter(s) to be tuned end-to-end via our proposed validation loss. Experiments on two testbeds with semantic class anomalies and subtle industrial defects show that a systematic tuning of augmentation gives significant performance gains over current practices. All our code and testbeds are available at https://anonymous.4open.science/r/ST-SSAD.

## 1 INTRODUCTION

Anomaly detection (AD) finds many applications in security, finance, and manufacturing, to name a few. Thanks to its popularity, the literature is abound with numerous detection techniques (Aggarwal, 2016), while deep neural network-based models have attracted the most attention recently (Pang et al., 2021). Especially for adversarial or dynamically-changing settings in which the anomalies are to be identified, it is important to design *unsupervised* techniques. While supervised detection can be employed for label-rich settings, unsupervised detection becomes critical to remain alert to emerging phenomena or the so-called "unknown unknowns".

Recently, self-supervised learning (SSL) has emerged as a promising paradigm that offers supervisory signals to real-world problems while avoiding the extensive cost of manual labeling, leading to great success in advancing NLP (Conneau et al., 2020; Brown et al., 2020) as well as computer vision tasks (Goyal et al., 2021; He et al., 2022). SSL has become particularly attractive for *unsupervised* tasks such as AD, where labeled data is either nonexistent, costly to obtain, or nontrivial to simulate in the face of unknown anomalies. For this reason, the literature has seen a recent surge of SSL-based AD (SSAD) techniques (Golan & El-Yaniv, 2018; Hendrycks et al., 2019; Bergman & Hoshen, 2020; Li et al., 2021; Sehwag et al., 2021; Qiu et al., 2021). The typical approach to SSAD involves incorporating self-generated *pseudo* anomalies into training, and then learning to separate those from the inliers. The pseudo anomalies are most often synthesized artificially by transforming inliers through a data augmentation function, such as masking, blurring, or rotation.

In this paper, we address a fundamental challenge with SSAD, to which recent works seem to have turned a blind eye: recognizing and tuning *augmentation as a hyperparameter*. As shown recently by Yoo et al. (2023), the choice of a data augmentation function, as well as its associated arguments such as the masking amount, blurring level, etc., have tremendous impact on detection performance. This may come across as no surprise, since the supervised learning community has integrated "data augmentation hyperparameters" into model selection (Cubuk et al., 2019; Ottoni et al., 2023). Meanwhile, there exists no such attempt in the literature on SSAD (!). Although model selection without

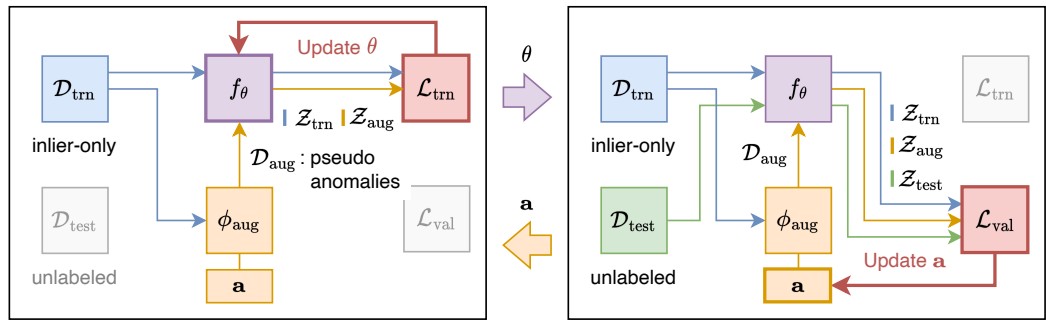

Figure 1: (left) Training and (right) validation (i.e. augmentation tuning) stages of ST-SSAD, which alternates between: (left) Given aug. function $\phi_{\text{aug}}(\cdot; \mathbf{a})$, estimate the parameters $\theta$ of detector $f_\theta$ on inliers $\mathcal{D}_{\text{trn}}$ and pseudo anomalies $\mathcal{D}_{\text{aug}}$ via the training loss; (right) Given $f_\theta$ and unlabeled test data $\mathcal{D}_{\text{test}}$, gradient-update the aug. hyperparameters $\mathbf{a}$ via our differentiable unsupervised validation loss that measures the agreement between $\mathcal{D}_{\text{trn}} \cup \mathcal{D}_{\text{aug}}$ and $\mathcal{D}_{\text{test}}$ in the embedding space.

ground-truth labels is admittedly a much harder problem, turning a blind eye to the challenge may mislead by overstating the (unreasonable) effectiveness of SSL for unsupervised AD.

Our work introduces the *first systematic approach for rigorous augmentation tuning on SSAD*. Intuitively, SSAD works well if the augmentation-generated pseudo anomalies resemble the true anomalies. Put differently, this is when the augmentation function well mimics the true anomaly-generating mechanism. Based on this insight, **(1)** we first design a novel, *unsupervised validation loss* for SSAD toward quantifying the alignment between the augmented training data and *unsupervised* test data. Then, we propose to tune augmentation through our differentiable validation loss *end-to-end* (see Fig. 1). This necessitates the augmentation function to be differentiable as well. To this end, **(2)** we propose new *differentiable formulations for popular augmentations* such as CutOut (local) (Devries & Taylor, 2017) and rotation (global) (Golan & El-Yaniv, 2018) as proof of concept.

We argue that the use of unlabeled test data, containing the anomalies to be identified, *during model tuning* transductively[1] is exceedingly important for the success of SSAD. This is fundamentally different from existing SSAD approaches that "imagine" how the actual anomalies would look like or otherwise haphazardly choose augmentation that corresponds to some arbitrary notion of anomalies, which may not well align with what is to be detected. One can incorporate expert or prior knowledge of anomalies in choosing augmentation, but in the absence thereof (recall unknown-unknowns), SSAD would likely fail as the recent study by Yoo et al. (2023) documents.

Our extensive experiments on 41 anomaly detection tasks including both local and global anomalies show that ST-SSAD significantly outperforms both unsupervised and self-supervised baselines which rely on manual hyperparameter search without labels. Our qualitative analysis visually supports that ST-SSAD is capable of learning appropriate augmentation hyperparameters for different anomaly types, even when they share the same normal data, by leveraging the anomalies in unlabeled test data. While we focus on image anomaly detection in this paper, our ST-SSAD framework is generally applicable to other data modalities given differentiable augmentation functions.

## 2 PRELIMINARIES

**Notation**  Let $\mathcal{D}_{\text{trn}}$ denote a set of training normal (i.e. inlier) data, and $\mathcal{D}_{\text{test}}$ be a set of test data containing both normal and anomalous samples. Let $\mathbf{x} \in \mathbb{R}^d$ denote a data sample in $\mathcal{D}_{\text{trn}} \cup \mathcal{D}_{\text{test}}$, where $d$ is its size. Let $\phi_{\text{aug}} \in \mathbb{R}^d \times \mathcal{A} \mapsto \mathbb{R}^d$ depict a data augmentation function conditioned on hyperparameters $\mathbf{a} \in \mathcal{A}$, where $\mathcal{A}$ is the set of possible values. For example, if $\phi_{\text{aug}}$ is the rotation of an image, $\mathcal{A} = [0, 360)$ is the domain of rotation angles, and $\phi_{\text{aug}}(\mathbf{x}; a)$ is the image rotated by angle $a$. Let $f_\theta \in \mathbb{R}^d \mapsto \mathbb{R}^h$ be a detector parameterized by $\theta$, and $s \in \mathbb{R}^h \mapsto \mathbb{R}^+$ be an anomaly score function. Specifically, $f_\theta$ returns a low-dimensional embedding $\mathbf{z} \in \mathbb{R}^h$ for each $\mathbf{x}$, which is then fed into $s$ to compute its anomaly score of $\mathbf{x}$. We assume that $f_\theta$ is trained in a self-supervised fashion by creating a set $\mathcal{D}_{\text{aug}} = \{\phi_{\text{aug}}(\mathbf{x}; \mathbf{a}) \mid \mathbf{x} \in \mathcal{D}_{\text{trn}}\}$ of pseudo anomalies using $\phi_{\text{aug}}$.

---

[1]Vapnik (2006) advocated transductive learning over inductive learning, stating that one should not solve a more general, harder problem, but rather solve the specific problem at hand directly. We argue that transduction is especially relevant for operationalizing SSL for AD.

**Problem definition** Given $\mathcal{D}_{\text{trn}}$ and $\mathcal{D}_{\text{test}}$, how can we find $\mathbf{a}^*$ (along with the model parameters $\theta$) that maximizes the accuracy of the detector $f_\theta$ with score function $s$?

There is no trivial solution to the problem, since the labeled anomalies are not given at training time. However, the problem is crucial for the success of SSAD in real-world tasks where the labels are hard to obtain or even nonexistent. To the best of our knowledge, this problem has not been studied in the literature, and our work is the first to propose a systematic solution to the problem.

## 3 PROPOSED FRAMEWORK FOR END-TO-END AUGMENTATION TUNING

We propose ST-SSAD (Self-Tuning Self-Supervised Anomaly Detection), a framework for augmentation hyperparameter tuning in SSAD. Given test data $\mathcal{D}_{\text{test}}$ which contains unlabeled anomalies, ST-SSAD automatically finds the best augmentation hyperparameter $\mathbf{a}^*$ that maximizes the semantic alignment between the augmentation function and the underlying anomaly-generating mechanism hidden in $\mathcal{D}_{\text{test}}$. The search process is performed in an end-to-end fashion thanks to two core novel engines of ST-SSAD: (1) an *unsupervised validation* loss $\mathcal{L}_{\text{val}}$ and (2) a *differentiable augmentation* function $\phi_{\text{aug}}$, which we describe in detail in Sec. 3.1 and 3.2, respectively.

Fig. 1 shows an overall structure of ST-SSAD, which updates the parameters $\theta$ of the detector $f_\theta$ and the augmentation hyperparameters $\mathbf{a}$ through alternating stages for training and validation. Let $\mathcal{Z}_{\text{trn}} = \{f_\theta(\mathbf{x}) \mid \mathbf{x} \in \mathcal{D}_{\text{trn}}\}$ be the embeddings of $\mathcal{D}_{\text{trn}}$, $\mathcal{Z}_{\text{aug}} = \{f_\theta(\phi_{\text{aug}}(\mathbf{x}; \mathbf{a})) \mid \mathbf{x} \in \mathcal{D}_{\text{trn}}\}$ be the embeddings of augmented data, and $\mathcal{Z}_{\text{test}} = \{f_\theta(\mathbf{x}) \mid \mathbf{x} \in \mathcal{D}_{\text{test}}\}$ be the embeddings of $\mathcal{D}_{\text{test}}$. In the training stage, ST-SSAD updates $\theta$ to minimize the training loss based on the pretext task of SSAD as determined by $\phi_{\text{aug}}$ with given $\mathbf{a}$. In the validation stage, ST-SSAD updates $\mathbf{a}$ to reduce the unsupervised validation loss based on the embeddings generated by the updated $f_\theta$. The framework halts when $\mathbf{a}$ reaches a local optimum, typically after a few iterations.

The detailed process of ST-SSAD is shown as Algo. 2 in Appendix A. Line 3 denotes the training stage, and Lines 4 to 8 represent the validation stage. $\theta$ is updated in Line 9 after the validation stage because of the second-order optimization (Sec. 3.3). Due to its gradient-based solution to the bilevel optimization problem, Algo. 2 is executed for multiple random initializations of $\mathbf{a}$ (Sec. 3.3).

### 3.1 UNSUPERVISED VALIDATION LOSS

The unsupervised validation loss $\mathcal{L}_{\text{val}}$ is one of the core components of ST-SSAD, which guides the direction of hyperparameter optimization. The goal is to quantify the agreement between $\phi_{\text{aug}}$ and the anomaly-generating mechanism. Our idea is to measure the set distance between $\mathcal{D}_{\text{trn}} \cup \mathcal{D}_{\text{aug}}$ and $\mathcal{D}_{\text{test}}$ in the embedding space, based on the intuition that the two sets will become similar, the more $\mathcal{D}_{\text{aug}}$ resembles the true anomalies in $\mathcal{D}_{\text{test}}$. Fig. 2

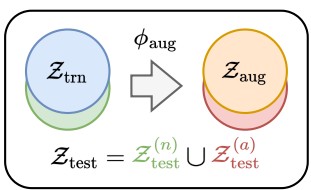

Figure 2: Illustration of $\mathcal{L}_{\text{val}}$.

depicts the intuition: we aim to find $\phi_{\text{aug}}$ that creates $\mathcal{Z}_{\text{aug}}$ similar to the set $\mathcal{Z}_{\text{test}}^{(a)}$ of true anomalies (in red), while matching $\mathcal{Z}_{\text{trn}}$ with the set $\mathcal{Z}_{\text{test}}^{(n)}$ of normal data (in green). By using the embeddings, we can avoid the high dimensionality of raw data and focus on their semantic representation.

Based on the idea, we present the basic form of our validation loss as follows:

$$\mathcal{L}_{\text{val}}^{(b)}(\mathcal{Z}_{\text{trn}}, \mathcal{Z}_{\text{aug}}, \mathcal{Z}_{\text{test}}) = \text{dist}(\mathcal{Z}_{\text{trn}} \cup \mathcal{Z}_{\text{aug}}, \mathcal{Z}_{\text{test}}), \tag{1}$$

where $\text{dist}(\cdot, \cdot)$ is a distance function between sets of vectors. The effectiveness of $\mathcal{L}_{\text{val}}^{(b)}$ is determined by how $\text{dist}$ is defined, which we carefully design to address two notable challenges:

- *Scale invariance:* During optimization, the scale of distances between embeddings can arbitrarily change as $\mathbf{a}$ is updated, which makes the value of $\mathcal{L}_{\text{val}}$ inconsistent. Thus, $\mathcal{L}_{\text{val}}$ should be robust to the scale of distances as long as the (relative) distribution of embeddings is preserved.
- *Ratio robustness:* Let $\gamma = |\mathcal{D}_{\text{aug}}|/|\mathcal{D}_{\text{trn}}|$ denote the relative size of augmented data, which means the number of times we apply $\phi_{\text{aug}}$ to $\mathcal{D}_{\text{trn}}$. Since the exact anomaly ratio in $\mathcal{D}_{\text{test}}$ is not known, $\mathcal{L}_{\text{val}}$ should be robust to the value of $\gamma$ which we manually set prior to training.

**Total distance normalization** For scale invariance, we propose total distance normalization to unify the total pairwise squared distance (TPSD) (Zhao & Akoglu, 2020) of embeddings. Let $\mathbf{Z}$ be an embedding matrix that stacks all embedding vectors in $\mathcal{Z}_{\text{trn}}$, $\mathcal{Z}_{\text{aug}}$, and $\mathcal{Z}_{\text{test}}$ as its rows. Then,

TPSD is defined as $\text{TPSD}(\mathbf{Z}) = \sum_{ij} \|\mathbf{z}_i - \mathbf{z}_j\|_2^2$, where $\mathbf{z}_i$ is the $i$-th row of $\mathbf{Z}$. We can transform any $\mathbf{Z}$ to have the unit TPSD in linear time (Zhao & Akoglu, 2020) via

$$\mathbf{z}_i' = \frac{\sqrt{N}}{\|\mathbf{Z}^c\|_{\text{F}}} \mathbf{z}_i^c \quad \text{where} \quad \mathbf{z}_i^c = \mathbf{z}_i - \frac{1}{N}\sum_{j=1}^N \mathbf{z}_j \;, \tag{2}$$

where $\|\cdot\|_{\text{F}}$ is the Frobenius norm of a matrix, and $N$ is the number of rows in $\mathbf{Z}$.

By using $\mathbf{Z}'$ instead of $\mathbf{Z}$ for computing the distances, we can focus on the relative distances between embeddings while maintaining the overall variance. It is noteworthy that the vector normalization, i.e., $\mathbf{z}_i \leftarrow \mathbf{z}_i/\|\mathbf{z}_i\|_2 \; \forall i$, does not solve the scale invariance problem since the scale of distances can still be arbitrary even on unit vectors. Another advantage that total distance normalization offers is that it steers away from the trivial solution, which is to set all embeddings to the zero vector.

**Mean distance loss** For ratio robustness, we use the asymmetric mean distance as the dist function to separate $\text{dist}(\mathcal{Z}_{\text{trn}} \cup \mathcal{Z}_{\text{aug}}, \mathcal{Z}_{\text{test}})$ into $(\text{dist}(\mathcal{Z}_{\text{trn}}, \mathcal{Z}_{\text{test}}) + \text{dist}(\mathcal{Z}_{\text{aug}}, \mathcal{Z}_{\text{test}}))/2$ as follows.

$$\mathcal{L}_{\text{val}}(\mathcal{Z}_{\text{trn}}, \mathcal{Z}_{\text{aug}}, \mathcal{Z}_{\text{test}}) = \frac{1}{2} \sum_{\mathbf{z}' \in \mathcal{Z}_{\text{test}}'} \|\mathbf{z}' - \text{mean}(\mathcal{Z}_{\text{trn}}')\|_2 + \|\mathbf{z}' - \text{mean}(\mathcal{Z}_{\text{aug}}')\|_2 \;, \tag{3}$$

where $\mathcal{Z}_{\text{trn}}'$, $\mathcal{Z}_{\text{aug}}'$, and $\mathcal{Z}_{\text{test}}'$ are the embeddings after the total distance normalization, and $\text{mean}(\cdot)$ is the (elementwise) mean of a set of vectors. The mean operation allows $\mathcal{L}_{\text{val}}$ to be invariant to the individual (or internal) distributions of $\mathcal{Z}_{\text{trn}}$ and $\mathcal{Z}_{\text{aug}}$, including their sizes, while focusing on their global relative positions with respect to $\mathcal{Z}_{\text{test}}$. This is another desired property for $\mathcal{L}_{\text{val}}$, since we want to avoid minimizing $\mathcal{L}_{\text{val}}$ only by decreasing the variance of $\mathcal{Z}_{\text{aug}}$.

**Theoretical properties** We study $\mathcal{L}_{\text{val}}$ theoretically on singleton scenarios, where the sets $\mathcal{Z}_{\text{trn}}$, $\mathcal{Z}_{\text{aug}}$, $\mathcal{Z}_{\text{test}}^{(n)}$, and $\mathcal{Z}_{\text{test}}^{(a)}$ are all of size one. We claim in Lemma 1 that $\mathcal{L}_{\text{val}}$ is one with the perfect alignment, and in Lemma 2 that we can reach the perfect alignment by minimizing $\mathcal{L}_{\text{val}}$ via gradient-based optimization. Fig. 3 provides empirical evidence on Lemma 2, where we assume that every embedding is a vector of length one. All the negative gradients point to the perfect alignment achieved when $u_1 = u_2 = 0$. Proofs are in Appendix B.



**Lemma 1** (Perfect alignment). $\mathcal{L}_{\text{val}}(\mathcal{Z}_{\text{trn}}, \mathcal{Z}_{\text{aug}}, \mathcal{Z}_{\text{test}}) = 1$ if $|\mathcal{Z}_{\text{trn}}| = |\mathcal{Z}_{\text{aug}}| = 1$, $\mathcal{Z}_{\text{trn}} = \mathcal{Z}_{\text{test}}^{(n)}$, and $\mathcal{Z}_{\text{aug}} = \mathcal{Z}_{\text{test}}^{(a)}$.

**Lemma 2** (Existence of local optima). Let $\mathcal{Z}_{\text{trn}} = \{\mathbf{z}_1\}$, $\mathcal{Z}_{\text{aug}} = \{\mathbf{z}_2\}$, $\mathcal{Z}_{\text{test}}^{(n)} = \{\mathbf{z}_1 + \mathbf{u}_1\}$, and $\mathcal{Z}_{\text{test}}^{(a)} = \{\mathbf{z}_2 + \mathbf{u}_2\}$. There exists $\delta > 0$ such that if $\|\mathbf{u}_1\|_2 \le \delta$ and $\|\mathbf{u}_2\|_2 \le \delta$, $\mathcal{L}_{\text{val}}(\mathcal{Z}_{\text{trn}}, \mathcal{Z}_{\text{aug}}, \mathcal{Z}_{\text{test}}) = 1$ if and only if $\mathbf{u}_1 = \mathbf{u}_2 = 0$.

Figure 3: $\mathcal{L}_{\text{val}}$ (as color) for different values of $u_1$ and $u_2$, and the negative gradients of $\mathcal{L}_{\text{val}}$ with respect to $u_1$ and $u_2$ (as arrows).

### 3.2 DIFFERENTIABLE AUGMENTATION

The second driving engine is a differentiable augmentation function that enables ST-SSAD to conduct end-to-end optimization. Some functions are inherently differentiable if implemented in a correct way, while others require differentiable surrogate functions that provide a similar functionality. As proof of concept, we take this approach to introduce two differentiable augmentation functions; one representative of local and another representative of global augmentation. Specifically, we propose CutDiff (§3.2.1) as a differentiable variant of CutOut (Devries & Taylor, 2017) that is originally designed for localized anomalies. We also utilize a differentiable formulation of Rotation (§3.2.2), which is a popular augmentation function which transforms the input globally and has been widely used for semantic anomaly detection (Bergman & Hoshen, 2020; Sehwag et al., 2021).

### 3.2.1 CUTDIFF FOR LOCAL AUGMENTATION

Local augmentation such as CutOut (Devries & Taylor, 2017) and CutPaste (Li et al., 2021) mimics subtle local anomalies by modifying a partial region in an image. CutOut removes a small patch from an image and fills in it with a black patch, while CutPaste copies a small patch and pastes it into a different random location of the same image. However, all of these functions are not differentiable, and thus cannot be directly used for our end-to-end ST-SSAD framework.

We propose CutDiff in Algo. 1, which creates a smooth round patch and extracts it from the given image in a differentiable way. The idea is to model the patch shape as a function of hyperparameters

---

**Algorithm 1** Proposed CutDiff Augmentation

---

**Input:** Image $\mathbf{x}$ and augmentation hyperparameters $\mathbf{a} \in \mathbb{R}^3$
**Output:** Augmented image $\tilde{\mathbf{x}}$
1: $\mathbf{X} \leftarrow$ Reshape $\mathbf{x}$ as a tensor of size $m \times m \times 3$, assuming an RGB image
2: $\mathbf{L} \leftarrow$ Reshape $\mathbf{a}$ as a lower triangular matrix of size $2 \times 2$
3: $\mathbf{G} \leftarrow$ Create a grid tensor of size $m \times m \times 2$ such that $\mathbf{g}_{ij} = (i/m, j/m) \; \forall i, j \in [1, m]$
4: $\boldsymbol{\mu} \leftarrow$ Sample a position vector of size 2 such that $\mu_1, \mu_2 \sim \text{uniform}(0, 1)$
5: $\mathbf{P} \leftarrow$ Create a patch matrix such that $p_{ij} = \exp(-(\mathbf{g}_{ij} - \boldsymbol{\mu})^\top (\mathbf{L}\mathbf{L}^\top)^{-1}(\mathbf{g}_{ij} - \boldsymbol{\mu}))$
6: $\tilde{\mathbf{x}} \leftarrow$ Reshape $\tilde{\mathbf{X}}$ where $\tilde{x}_{ijk} = \min(\max(x_{ijk} - p_{ij}, 0), 1) \; \forall i, j, k$

---

$\mathbf{a} \in \mathbb{R}^3$ by computing the strength of the patch based on the distance between the patch center and each image pixel. By reshaping $\mathbf{a}$ as a lower triangular matrix $\mathbf{L}$ and changing it, we in effect tune the rotated angle, size, and ratio of the patch. In Appendix C, we provide visualizations of CutDiff compared with CutOut and CutPaste, as well as its implementation details.

### 3.2.2 Rotation for Global Augmentation

Geometric augmentation such as rotation, translation, and flipping has been widely used for image anomaly detection (Golan & El-Yaniv, 2018; Bergman & Hoshen, 2020). Unlike local augmentation such as CutOut, many geometric transformations are differentiable if they are represented as matrix-vector operations. We adopt the differentiable image rotation function proposed by Jaderberg et al. (2015), which consists of two main steps. First step is the creation of a $2 \times 3$ rotation matrix. Second step is to create a sampling function that selects a proper pixel position of the given image for each position of the target image based on the rotation matrix and the affine grid. The resulting operation is differentiable, since it is a parameterized sampling between pixels of the two images.

### 3.3 Techniques for Practical Usability

We introduce two additional techniques for improving the practical usability and generalizability of ST-SSAD toward real-world data: *second-order optimization* and *multiple initialization*.

**Second-order optimization**  At each training iteration, ST-SSAD updates augmentation hyperparameters $\mathbf{a}$ and the parameters $\theta$ of the detection network $f_\theta$ through alternating stages. We expect the following inequality to hold:
$$\mathcal{L}_{\text{val}}(\mathbf{a}^{(t+1)}, \theta') \; < \; \mathcal{L}_{\text{val}}(\mathbf{a}^{(t)}, \theta) \; , \tag{4}$$
where $t$ is the current iteration, and $\theta'$ denotes the updated parameters of the detector $f_\theta$ derived by using $\mathbf{a}^{(t)}$ to generate pseudo anomalies for its training.

However, the first-order optimization of $\mathbf{a}$ cannot take into account that the parameters $\theta'$ and $\theta$ are different between both sides of Eq. 4, as it treats $\theta'$ as a constant. As a solution, ST-SSAD considers $\theta'$ as a function of $\mathbf{a}^{(t)}$ and conducts second-order optimization as follows:
$$\mathbf{a}^{(t+1)} = \mathbf{a}^{(t)} - \beta \nabla_{\mathbf{a}^{(t)}} \mathcal{L}_{\text{val}}(\mathbf{a}^{(t)}, \theta - \alpha \nabla_\theta \mathcal{L}_{\text{trn}}(\theta, \mathbf{a}^{(t)})) \; . \tag{5}$$

In this way, the optimization process can accurately track the change in $\theta$ caused by the update of $\mathbf{a}$, resulting in a stable minimization of $\mathcal{L}_{\text{val}}$. Note that Eq. 5 is the same as in Line 8 of Algorithm 2, except we assume that $\mathcal{L}_{\text{val}}$ takes $(\mathbf{a}, \theta)$ as its inputs in Eq. 4 and 5.

**Multiple initialization**  The result of ST-SSAD is affected by how we initialize the augmentation hyperparameters $\mathbf{a}$, since it conducts gradient-based updates toward local optima. A natural way to address initialization is to pick a few random starting points and select the best one. However, it is difficult to fairly select the best from multiple points, since $\mathcal{L}_{\text{val}}$ is designed to locally improve the current $\mathbf{a}$, rather than to compare different models; e.g., it is possible that a less-aligned model can produce lower $\mathcal{L}_{\text{val}}$ if the augmented data are distributed more sparsely in the embedding space.

As a solution, we propose a simple yet effective measure to enable the comparison between models from different initialization choices. Let $s$ be the anomaly score function as presented in the problem definition in Sec. 2. Then, we define the score variance of the test data as
$$\mathcal{S}(\theta) = \frac{\sum_{s \in \mathcal{C}} (s - \text{mean}(\mathcal{C}))^2}{|\mathcal{D}_{\text{test}}| - 1} \quad \text{where} \quad \mathcal{C} = \{s(f_\theta(\mathbf{x})) \mid \mathbf{x} \in \mathcal{D}_{\text{test}}\}. \tag{6}$$
We use $\mathcal{S}$ (the larger, the better) to select the best initialization point after training completes. The idea is that the variance of test anomaly scores is likely to be large under a good augmentation, as it

generally reflects a better separability between inliers and anomalies in the test data, and it offers a fair evaluation since ST-SSAD does not observe the score function $s$ at all during optimization.

# 4 EXPERIMENTS

**Datasets** We run experiments on 41 different anomaly detection tasks, which include 23 subtle (local) anomalies in MVTec AD (Bergmann et al., 2019) and 18 semantic (gross) anomalies in SVHN (Netzer et al., 2011). MVTec AD is an image dataset of industrial objects, where the anomalies are local defects such as scratches. We include four types of objects in our experiments: Cable, Carpet, Grid, and Tile, each of which contains five to eight anomaly types. SVHN is a digits image dataset from house numbers in Google Street View. We use digits 2 and 6 as normal classes and treat the remaining digits as anomalies, generating 18 different tasks for all pairs of digits (2 vs. others and 6 vs. others). Note that our experimental setup is different from previous works (Li et al., 2021), since we focus on the performance on each anomaly type rather than overall accuracy.

**Model settings** We use a detector network $f_\theta$ with the same ResNet-based architecture (He et al., 2016) as in a previous work (Li et al., 2021). We use binary cross entropy as the training loss $\mathcal{L}_{\text{trn}}$ for classification between normal and augmented data, applying an MLP head to the embeddings to produce prediction outputs. As the anomaly score function $s$, we use the negative log likelihood of a Gaussian density estimator learned on the embeddings of training data as in previous works (Rippel et al., 2020; Li et al., 2021). For ST-SSAD, we use four initialization points for each augmentation function: $\{0.0001, 0.001, 0.01, 0.1\}$ for the CutDiff patch size, and $\{45°, 135°, 225°, 315°\}$ for the Rotation angle. We set both the initial patch angle and ratio to zero. We employ CutDiff and Rotation for defect and semantic anomaly detection tasks, respectively. The sum of training and validation losses is used as the stopping criterion for the updates to hyperparameters $\mathbf{a}$.

**Evaluation metrics** The accuracy of each model is measured by the area under the ROC curve (AUC) on the anomaly scores computed for $\mathcal{D}_{\text{test}}$. We run all experiments five times and report the average and standard deviation. For statistical comparison between different models on all tasks and random seeds, we also run the paired Wilcoxon signed-rank test (Groggel, 2000). The one-sided test with $p$-values smaller than $0.05$ represents that our ST-SSAD is statistically better than the other.

**Baselines** To the best of our knowledge, there are no direct competitors on end-to-end augmentation hyperparameter tuning for SSAD. Thus, we compare ST-SSAD with various types of baselines: *SSL without hyperparameter tuning*—(1) random dynamic selection (RD) that selects $\mathbf{a}$ randomly at each training epoch, and (2) random static selection (RS) that selects $\mathbf{a}$ once before the training begins. *Unsupervised learning*—(3) autoencoder (AE) (Golan & El-Yaniv, 2018) and (4) DeepSVDD (Ruff et al., 2018). *Variants of our ST-SSAD with naïve choices*—(5) using maximum mean discrepancy (MMD) (Gretton et al., 2006) as $\mathcal{L}_{\text{val}}$, (6) MMD without the total distance normalization, and (7) using first-order optimization. RD and RS are used with either of CutOut, CutPaste, CutDiff, or Rotation, which we denote CO, CP, CD, and RO, respectively, for brevity. We also denote baselines (5)–(7) as MMD1, MMD2, and FO, respectively. Additional details are in Appendix D.

## 4.1 DEMONSTRATIVE EXAMPLES

We first present experimental results on demonstrative datasets, where we create synthetic anomalies through CutDiff. Given normal images of the Carpet object in the MVTec AD dataset, we generate 25 types of anomalies with the patch size in $\{0.01, 0.02, 0.04, 0.08, 0.16\}$ and the aspect ratio in $\{0.25, 0.5, 1.0, 2.0, 4.0\}$, where the angle is fixed to 0. Our goal is to demonstrate that ST-SSAD is able to learn different $\mathbf{a}$ for different anomaly types in these controlled settings.

Fig. 4 shows the experimental results. In Fig. 4a, ST-SSAD learns different values of $\mathbf{a}$ depending on the properties of anomalies, patch size and patch ratio, demonstrating the ability of ST-SSAD to adapt to varying anomalies. Nevertheless, there exists a slight difference between the learned $\mathbf{a}$ and the true values in some cases, since embedding distributions can be matched as in Fig. 4b even with a small difference. This difference is typically larger for patch ratio than for patch size, suggesting that patch size impacts the embeddings more than the ratio does. Fig. 4c depicts the training process of ST-SSAD for five anomaly types generated with different patch sizes, where the patch ratio is set to 1.0. We visualize the average and standard deviation from five runs with different random seeds. ST-SSAD accurately adapts to different patch sizes even from the same initialization point, updating $\mathbf{a}$ through iterations to minimize the validation loss.

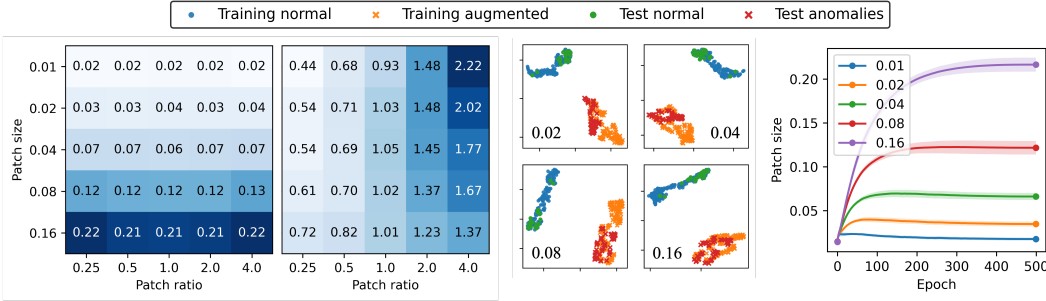

(a) Learned values of patch size (left) & ratio (right)    (b) $t$-SNE embeddings    (c) Patch size during training

Figure 4: Experimental results on demonstrative examples, where we create 25 types of anomalies using CutDiff with five different patch size and patch ratio values, respectively. (a) Each cell's value represents what ST-SSAD has learned through end-to-end optimization for the (left) patch size and (right) ratio, respectively. ST-SSAD successfully mimics the true values of the patch ratio and size shown in the $x$- and $y$-axes, respectively, until (b) the embedding distributions are matched between $\mathcal{Z}_{\mathrm{trn}} \cup \mathcal{Z}_{\mathrm{aug}}$ and $\mathcal{Z}_{\mathrm{test}}$, showing a learning trajectory like (c). AUC is $1.00$ in all 25 tasks.

## 4.2 Testbed Evaluation

Next, we perform quantitative evaluation of ST-SSAD on both industrial-defect anomalies and semantic class anomalies, covering 41 different anomaly detection tasks. Table 1 provides the results on 23 tasks for industrial-defect anomalies. ST-SSAD (1) achieves the best AUC in 9 different tasks, and (2) outperforms 7 of the 8 baselines with $p$-values smaller than $0.01$, showing strong statistical significance throughout all 23 tasks. The $p$-value is still small for the remaining case ($0.07$) when compared to RD-CD, which means that ST-SSAD outperforms RD-CD in most tasks in the testbed. Table 2 shows the experimental results on 18 tasks for semantic class anomalies, where ST-SSAD significantly outperforms all baselines with all $p$-values smaller than $0.0001$.

The ablation studies between ST-SSAD and its three variants in Tables 1 and 2 show the effectiveness of our ideas that compose ST-SSAD: total distance normalization, mean distance loss, and second-order optimization. Especially, the two MMD-based methods are significantly worse than our ST-SSAD, showing that MMD is not suitable for augmentation tuning even though it is widely used as a set distance measure, due to the challenges we aim to address with $\mathcal{L}_{\mathrm{val}}$. The difference between ST-SSAD and the first-order baseline is smaller, meaning that the first-order optimization can still be used for ST-SSAD when the computational efficiency needs to be prioritized.

## 4.3 Qualitative Analysis

We also perform qualitative analysis as shown in Fig. 6 and 7, visualizing the augmentation functions learned by ST-SSAD for different types of anomalies. Fig. 6 illustrates three types of anomalies in the Cable object and one type of anomaly in the Carpet object. The four anomaly types have their own sizes and aspect ratios of defected regions, which are accurately learned by ST-SSAD. Note that the three types of Cable anomalies share the training data $\mathcal{D}_{\mathrm{trn}}$; ST-SSAD captures their difference only from the alignment. The locations of patches created by CutDiff are chosen randomly at each run, since the locations of local defects are different for each anomalous image.

Fig. 7 illustrates images in the SVHN dataset and the embedding distributions after the training of ST-SSAD is completed. Fig. 7a and 7c show that ST-SSAD learns $180°$ as the angle of Rotation as $\phi_{\mathrm{aug}}$, since the anomalies in both tasks can be resembled by the $180°$-rotated normal images. After the training, the embedding distributions between $\mathcal{Z}_{\mathrm{trn}} \cup \mathcal{Z}_{\mathrm{aug}}$ and $\mathcal{Z}_{\mathrm{test}}$ are matched as shown in Fig. 7b and 7d, achieving high average AUC of $0.944$ and $0.887$, respectively (see Table 2).

## 4.4 Discussion

Table 1 shows the success of ST-SSAD throughout different tasks, but it also implies that ST-SSAD cannot always improve detection accuracy in all tasks. In some tasks like Rough anomalies in Tile, a simple baseline like random CutOut shows higher AUC than other models. This is because some anomaly types are difficult to mimic with CutDiff due to the inherent mismatch of the augmentation function. Fig. 5 shows two example anomaly types, Tile-Oil and Carpet-Thread, where ST-SSAD cannot improve over the baselines. Local defects of Oil are brighter than the background, whereas

Table 1: Test AUC on 23 different tasks for subtle anomaly detection. Each number is the average from five runs, and the best in each row is in bold. ST-SSAD outperforms most baselines, which is supported by the $p$-values in the last row derived from the Wilcoxon signed rank test.

| Object | Anomaly Type | AE | D-SVDD | RS-CO | RD-CO | RS-CP | RD-CP | RS-CD | RD-CD | ST-SSAD | MMD1 | MMD2 | FO |
|---|---|---|---|---|---|---|---|---|---|---|---|---|---|
| | | | | | Main Result | | | | | | | Ablation Study | |
| Cable | Bent wire | 0.515 | 0.432 | 0.556 | 0.560 | 0.703 | **0.756** | 0.527 | 0.580 | 0.490 | 0.581 | 0.643 | 0.579 |
| Cable | Cable swap | 0.639 | 0.295 | 0.483 | 0.625 | 0.618 | 0.683 | 0.574 | **0.696** | 0.532 | 0.510 | 0.562 | 0.545 |
| Cable | Combined | 0.584 | 0.587 | 0.879 | 0.857 | 0.880 | **0.949** | 0.901 | 0.879 | 0.925 | 0.939 | 0.962 | 0.882 |
| Cable | Cut inner insulation | 0.758 | 0.591 | 0.630 | 0.737 | 0.766 | **0.833** | 0.623 | 0.732 | 0.667 | 0.633 | 0.649 | 0.689 |
| Cable | Cut outer insulation | **0.989** | 0.343 | 0.695 | 0.815 | 0.787 | 0.871 | 0.703 | 0.790 | 0.516 | 0.428 | 0.461 | 0.527 |
| Cable | Missing cable | 0.920 | 0.466 | 0.953 | 0.961 | 0.755 | 0.801 | 0.935 | 0.945 | **0.998** | 0.855 | 0.772 | 0.999 |
| Cable | Missing wire | 0.433 | 0.494 | 0.781 | 0.655 | 0.501 | 0.546 | 0.708 | 0.620 | **0.863** | 0.547 | 0.477 | 0.699 |
| Cable | Poke insulation | 0.287 | 0.471 | 0.469 | 0.527 | 0.645 | **0.672** | 0.489 | 0.503 | 0.630 | 0.692 | 0.816 | 0.676 |
| Carpet | Color | 0.578 | 0.716 | 0.669 | 0.508 | 0.412 | 0.287 | 0.643 | 0.639 | **0.938** | 0.761 | 0.741 | 0.918 |
| Carpet | Cut | 0.198 | 0.758 | 0.439 | 0.608 | 0.403 | 0.411 | 0.490 | 0.767 | **0.790** | 0.353 | 0.401 | 0.595 |
| Carpet | Hole | 0.626 | 0.676 | 0.379 | 0.613 | 0.404 | 0.389 | 0.470 | **0.765** | 0.590 | 0.438 | 0.229 | 0.630 |
| Carpet | Metal contamination | 0.056 | **0.739** | 0.198 | 0.304 | 0.240 | 0.167 | 0.255 | 0.474 | 0.076 | 0.392 | 0.134 | 0.392 |
| Carpet | Thread | 0.394 | **0.742** | 0.494 | 0.585 | 0.469 | 0.517 | 0.508 | 0.679 | 0.483 | 0.492 | 0.541 | 0.642 |
| Grid | Bent | **0.849** | 0.168 | 0.456 | 0.322 | 0.421 | 0.433 | 0.337 | 0.354 | 0.771 | 0.780 | 0.650 | 0.602 |
| Grid | Broken | 0.806 | 0.183 | 0.397 | 0.312 | 0.487 | 0.502 | 0.340 | 0.392 | **0.869** | 0.845 | 0.887 | 0.884 |
| Grid | Glue | 0.704 | 0.143 | 0.634 | 0.568 | 0.674 | 0.732 | 0.681 | 0.578 | **0.906** | 0.966 | 0.974 | 0.721 |
| Grid | Metal contamination | 0.851 | 0.229 | 0.421 | 0.380 | 0.499 | 0.514 | 0.425 | 0.613 | **0.858** | 0.861 | 0.665 | 0.732 |
| Grid | Thread | 0.583 | 0.209 | 0.612 | 0.494 | 0.500 | 0.549 | 0.654 | 0.611 | **0.973** | 0.962 | 0.969 | 0.964 |
| Tile | Crack | 0.770 | 0.728 | 0.872 | 0.993 | 0.743 | 0.636 | 0.837 | **0.999** | 0.749 | 0.740 | 0.820 | 0.595 |
| Tile | Glue strip | 0.697 | 0.509 | 0.693 | **0.836** | 0.665 | 0.700 | 0.675 | 0.831 | 0.767 | 0.585 | 0.649 | 0.561 |
| Tile | Gray stroke | 0.637 | 0.785 | 0.845 | 0.642 | 0.583 | 0.657 | 0.856 | 0.802 | **0.974** | 0.653 | 0.706 | 0.973 |
| Tile | Oil | 0.414 | 0.690 | 0.708 | 0.745 | 0.464 | 0.576 | 0.683 | **0.837** | 0.554 | 0.548 | 0.614 | 0.555 |
| Tile | Rough | 0.724 | 0.387 | 0.606 | **0.725** | 0.631 | 0.661 | 0.568 | 0.657 | 0.690 | 0.700 | 0.549 | 0.605 |
| | $p$-value | .0000 | .0000 | .0000 | .0012 | .0000 | .0000 | .0000 | .0728 | **Ours** | .0268 | .0073 | .1332 |

Table 2: Test AUC on 18 different tasks for semantic anomaly detection. The format is the same as in Table 1. ST-SSAD outperforms all baselines with the $p$-values smaller than 0.0001 in all cases.

| Object | Anomaly | AE | D-SVDD | RS-RO | RD-RO | ST-SSAD | MMD1 | MMD2 | FO |
|---|---|---|---|---|---|---|---|---|---|
| | | | | Main Result | | | | Ablation Study | |
| Digit 2 | Digit 0 | 0.602 | 0.472 | 0.672 | 0.734 | **0.816** | 0.519 | 0.518 | 0.506 |
| Digit 2 | Digit 1 | 0.544 | 0.499 | 0.601 | 0.609 | **0.743** | 0.499 | 0.501 | 0.498 |
| Digit 2 | Digit 3 | 0.604 | 0.503 | 0.664 | 0.730 | **0.832** | 0.508 | 0.510 | 0.511 |
| Digit 2 | Digit 4 | 0.561 | 0.511 | 0.679 | 0.746 | **0.790** | 0.531 | 0.530 | 0.530 |
| Digit 2 | Digit 5 | 0.625 | 0.502 | 0.709 | 0.824 | **0.877** | 0.512 | 0.514 | 0.517 |
| Digit 2 | Digit 6 | 0.616 | 0.496 | 0.726 | 0.826 | **0.887** | 0.511 | 0.507 | 0.510 |
| Digit 2 | Digit 7 | 0.541 | 0.496 | 0.584 | 0.639 | **0.823** | 0.521 | 0.520 | 0.518 |
| Digit 2 | Digit 8 | 0.616 | 0.498 | 0.673 | 0.738 | **0.805** | 0.524 | 0.524 | 0.522 |
| Digit 2 | Digit 9 | 0.588 | 0.485 | 0.625 | **0.687** | 0.659 | 0.516 | 0.523 | 0.518 |
| Digit 6 | Digit 0 | 0.531 | 0.480 | 0.586 | 0.606 | **0.777** | 0.503 | 0.514 | 0.504 |
| Digit 6 | Digit 1 | 0.517 | 0.498 | 0.621 | 0.610 | **0.854** | 0.516 | 0.528 | 0.511 |
| Digit 6 | Digit 2 | 0.594 | 0.503 | 0.735 | 0.807 | **0.916** | 0.525 | 0.531 | 0.529 |
| Digit 6 | Digit 3 | 0.570 | 0.507 | 0.686 | 0.750 | **0.823** | 0.518 | 0.520 | 0.521 |
| Digit 6 | Digit 4 | 0.525 | 0.508 | 0.659 | 0.703 | **0.709** | 0.527 | 0.521 | 0.530 |
| Digit 6 | Digit 5 | 0.544 | 0.502 | 0.615 | **0.661** | 0.658 | 0.516 | 0.513 | 0.509 |
| Digit 6 | Digit 7 | 0.567 | 0.505 | 0.699 | 0.729 | **0.861** | 0.540 | 0.551 | 0.541 |
| Digit 6 | Digit 8 | 0.546 | 0.500 | 0.575 | 0.641 | **0.732** | 0.512 | 0.523 | 0.512 |
| Digit 6 | Digit 9 | 0.579 | 0.495 | 0.708 | 0.817 | **0.944** | 0.527 | 0.519 | 0.524 |
| | $p$-value | .0000 | .0000 | .0000 | .0000 | **Ours** | .0000 | .0000 | .0000 |

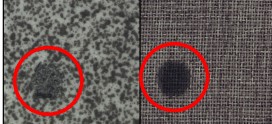
(a) Can be found by CutDiff

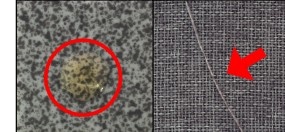
(b) Difficult with CutDiff

Figure 5: Some anomalies are hard to detect by CutDiff due to the inherent mismatch. See Sec. 4.4 for discussion.

CutDiff rather darkens the chosen patch. Anomalies of Thread type contain long thin threads, which are also hard to represent with CutDiff regardless of hyperparameter values.

ST-SSAD is a general framework, rather than a specific method, and its performance is affected by the detector model and augmentation function used. As the first systematic study for unsupervised augmentation tuning, we propose two differentiable augmentation functions for local and semantic anomalies, respectively, and demonstrate the success of ST-SSAD on two types of testbeds as proof of concept. We leave it as a future work to design a broader family of differentiable augmentations which can deal with more diverse types of anomalies, that also apply to other data modalities.

## 5 RELATED WORK

**Self-supervised learning (SSL)** has seen a surge of attention especially for pre-training foundation models (Bommasani et al., 2021), like LLMs which can generate remarkable human-like text (Zhou et al., 2023). Self-supervised representation learning has also offered astonishing boost to a variety of tasks in NLP, vision, and recommender systems (Liu et al., 2021). In fact, SSL has been argued as the key toward "unlocking the dark matter of intelligence" (LeCun & Misra, 2021).

**Self-supervised anomaly detection (SSAD)**: In terms of the pretext task and the associated loss on which they are trained, most SSL methods can be categorized as generative or contrastive. Genera-

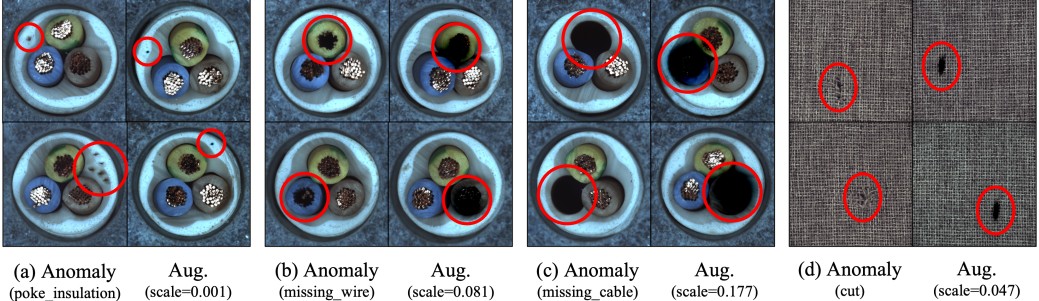

Figure 6: Illustrations of four anomaly types for the Cable and Carpet objects and the corresponding augmentations learned by ST-SSAD. Different hyperparameters of CutDiff are learned to resemble the true anomalies, including both the size and the aspect ratio of a patch.

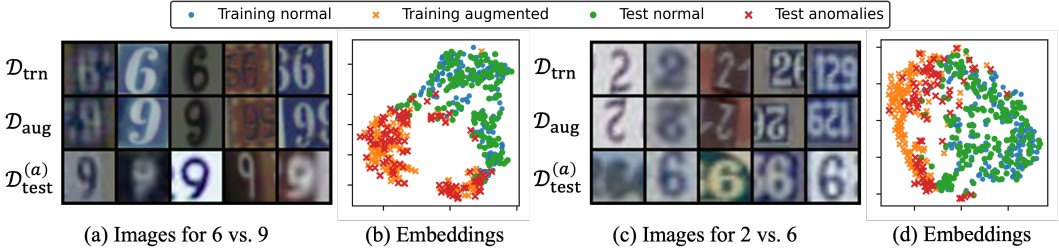

Figure 7: Illustrations of learned augmentations on the SVHN dataset and the corresponding distributions of embeddings. The three rows in (a, c) represent normal images, augmented images, and anomalies, respectively. ST-SSAD successfully learns the rotation of $180°$ for both tasks, achieving a match also visually between the distributions of $\mathcal{Z}_{\text{trn}} \cup \mathcal{Z}_{\text{aug}}$ and $\mathcal{Z}_{\text{test}}$ as shown in (b, d).

tive SSAD can further be organized based on (denoising) autoencoders (Zhou & Paffenroth, 2017; Zong et al., 2018; Cheng et al., 2021; Ye et al., 2022), adversarial approaches (Akcay et al., 2018; Zenati et al., 2018), and flow-based models (Rudolph et al., 2022; Gudovskiy et al., 2022). On the other hand, contrastive SSAD relies on data augmentation that generates pseudo anomalies by transforming inliers, and a supervised loss for distinguishing between inliers and the pseudo anomalies (Hojjati et al., 2022). Augmentation strategies for contrastive SSAD include geometric (Golan & El-Yaniv, 2018; Bergman & Hoshen, 2020), localized cut-paste (Li et al., 2021), patch-wise cloning Schlüter et al. (2021), masking (Cho et al., 2021), distribution-shifting transformation (Tack et al., 2020), and learnable neural network-based transformation (Qiu et al., 2021).

**Automating augmentation:** Recent work in computer vision (CV) has shown that the success of SSL relies on well-designed data augmentation functions (Steiner et al., 2022; Touvron et al., 2022). Sensitivity to the choice of augmentation has also been shown for SSAD recently (Yoo et al., 2023). While data augmentation in CV aims at improving generalization by accounting for invariances (e.g. mirror reflection of a dog is still a dog), augmentation in SSAD plays the key role of presenting the classifier with specific kinds of pseudo anomalies. While the supervised CV community proposed methods toward automating augmentation (Cubuk et al., 2019; 2020), our proposed work is the first attempt toward rigorously tuning data augmentation for SSAD. The key difference is that the former sets aside a *labeled validation* set to measure generalization, whereas we address the arguably more challenging setting for *fully unsupervised* anomaly detection without any labels.

## 6 CONCLUSION

Our work presented ST-SSAD, the first framework for self-tuning self-supervised anomaly detection, which automatically tunes the augmentation hyperparameters in an end-to-end fashion. To this end, we addressed two key challenges: unsupervised validation and differentiable augmentation. We proposed a smooth validation loss that quantifies the agreement between augmented and test data in a tranductive fashion. We also introduced two differentiable formulations for both local and global augmentation, while ST-SSAD can flexibly accommodate any other differentiable augmentation. Our experiments on two large testbeds validated the superiority of ST-SSAD over existing practices. Future work will design differentiable formulations for other augmentation families and then also incorporate the discrete selection of augmentation as part of self-tuning for SSAD.

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

---

**Algorithm 2** ST-SSAD: Self-Tuning Self-Supervised Anomaly Detection

---

**Input:** Training data $\mathcal{D}_{\text{trn}}$, test data $\mathcal{D}_{\text{test}}$, augmentation function $\phi_{\text{aug}}$, training loss $\mathcal{L}_{\text{trn}}$, validation loss $\mathcal{L}_{\text{val}}$, detector $f_\theta$ with parameters $\theta$, number of epochs $T$, and step sizes $\alpha$ and $\beta$

**Output:** Optimized augmentation hyperparameters $\mathbf{a}^{(T)}$

1: $\mathbf{a}^{(0)} \leftarrow$ Initialize augmentation hyperparameters
2: **for** $t \in \{0, 1, \cdots, T-1\}$ **do**
3:      Let $\theta' = \theta - \alpha \nabla_\theta \mathcal{L}_{\text{trn}}(\theta, \mathbf{a}^{(t)})$          ▷ Get new parameters as a function of $\mathbf{a}^{(t)}$
4:      $\mathcal{Z}_{\text{trn}} \leftarrow \{f_{\theta'}(\mathbf{x}) \mid \mathbf{x} \in \mathcal{D}_{\text{trn}}\}$          ▷ Generate training embeddings
5:      $\mathcal{Z}_{\text{aug}} \leftarrow \{f_{\theta'}(\phi_{\text{aug}}(\mathbf{x}; \mathbf{a}^{(t)})) \mid \mathbf{x} \in \mathcal{D}_{\text{trn}}\}$      ▷ Generate augmented embeddings
6:      $\mathcal{Z}_{\text{test}} \leftarrow \{f_{\theta'}(\mathbf{x}) \mid \mathbf{x} \in \mathcal{D}_{\text{test}}\}$          ▷ Generate test embeddings
7:      $\mathcal{Z}_{\text{trn}}, \mathcal{Z}_{\text{aug}}, \mathcal{Z}_{\text{test}} \leftarrow \text{normalize}(\mathcal{Z}_{\text{trn}}, \mathcal{Z}_{\text{aug}}, \mathcal{Z}_{\text{test}})$      ▷ Normalize embeddings as in Eq. 2
8:      $\mathbf{a}^{(t+1)} \leftarrow \mathbf{a}^{(t)} - \beta \nabla_{\mathbf{a}} \mathcal{L}_{\text{val}}(\mathcal{Z}_{\text{trn}}, \mathcal{Z}_{\text{aug}}, \mathcal{Z}_{\text{test}})$    ▷ Update the augmentation hyperparameters
9:      $\theta \leftarrow \theta'$          ▷ Update the detector network parameters
10: **end for**

---

## A    ST-SSAD ALGORITHM

The detailed process of ST-SSAD is shown as Algo. 2. Line 3 denotes the training stage, and Lines 4 to 8 represent the validation stage. $\theta$ is updated in Line 9 after the validation stage because of the second-order optimization (Sec. 3.3). Due to its gradient-based solution to the bilevel optimization problem, Algo. 2 is executed for multiple random initializations of $\mathbf{a}$ (Sec. 3.3).

## B    PROOFS OF LEMMAS ON THE VALIDATION LOSS

### B.1    PROOF OF LEMMA 1

*Proof.* Let $\mathcal{Z}_{\text{trn}} = \mathcal{Z}_{\text{test}}^{(n)} = \{\mathbf{z}_1\}$ and $\mathcal{Z}_{\text{aug}} = \mathcal{Z}_{\text{test}}^{(a)} = \{\mathbf{z}_2\}$. Then, the embedding matrix $\mathbf{Z} \in \mathbb{R}^{h \times 4}$ before the total distance normalization is given as

$$\mathbf{Z} = \begin{bmatrix} \mathbf{z}_1 & \mathbf{z}_1 & \mathbf{z}_2 & \mathbf{z}_2 \end{bmatrix}.$$

Let $\bar{\mathbf{z}} = (\mathbf{z}_1 + \mathbf{z}_2)/2$ be the center of the two vectors, and $\bar{\mathbf{Z}} \in \mathbb{R}^{4 \times h}$ be a matrix where each column is $\bar{\mathbf{z}}$. Then, $\mathbf{Z}$ is transformed into $\tilde{\mathbf{Z}}$ as a result of the normalization:

$$
\begin{aligned}
\tilde{\mathbf{Z}} &= \frac{2}{\sqrt{2\sum_i (z_{1i} - \bar{z}_i)^2 + 2\sum_i (z_{2i} - \bar{z}_i)^2}} (\mathbf{Z} - \bar{\mathbf{Z}}) \\
&= \frac{2}{\sqrt{2\sum_i ((z_{1i} - \bar{z}_i)^2 + (z_{2i} - \bar{z}_i)^2)}} (\mathbf{Z} - \bar{\mathbf{Z}}) \\
&= \frac{2}{\sqrt{2\sum_i ((z_{1i} - z_{2i})^2/4 + (z_{2i} - z_{1i})^2/4)}} (\mathbf{Z} - \bar{\mathbf{Z}}) \\
&= \frac{2}{\sqrt{\sum_i (z_{1i} - z_{2i})^2}} (\mathbf{Z} - \bar{\mathbf{Z}}) \\
&= \frac{2}{\|\mathbf{z}_1 - \mathbf{z}_2\|_2} (\mathbf{Z} - \bar{\mathbf{Z}}).
\end{aligned}
$$

The validation loss $\mathcal{L}_{\text{val}}$ is computed on $\tilde{\mathbf{Z}}$ as follows:

$$
\begin{aligned}
\mathcal{L}_{\text{val}}(\mathcal{Z}_{\text{trn}}, \mathcal{Z}_{\text{aug}}, \mathcal{Z}_{\text{test}}) &= \frac{1}{4} \left( \|\tilde{\mathbf{z}}_1 - \tilde{\mathbf{z}}_1\|_2 + \|\tilde{\mathbf{z}}_1 - \tilde{\mathbf{z}}_2\|_2 + \|\tilde{\mathbf{z}}_2 - \tilde{\mathbf{z}}_1\|_2 + \|\tilde{\mathbf{z}}_2 - \tilde{\mathbf{z}}_2\|_2 \right) \\
&= \frac{1}{2} \|\tilde{\mathbf{z}}_1 - \tilde{\mathbf{z}}_2\|_2 \\
&= \frac{1}{2} \left\| \frac{2}{\|\mathbf{z}_1 - \mathbf{z}_2\|_2} ((\mathbf{z}_1 - \bar{\mathbf{z}}) - (\mathbf{z}_2 - \bar{\mathbf{z}})) \right\|_2 \\
&= 1.
\end{aligned}
$$

As a result, the lemma is proved.          $\square$

### B.2 PROOF OF LEMMA 2

*Proof.* Without loss of generality, we assume the simplest scalar embeddings of size one as $\mathcal{Z}_{\text{trn}} = \{0\}$, $\mathcal{Z}_{\text{test}}^{(n)} = \{u_1\}$, $\mathcal{Z}_{\text{aug}} = \{2\}$, $\mathcal{Z}_{\text{test}}^{(a)} = \{u_2 + 2\}$. Then, we show that $\mathcal{L}_{\text{val}}(\mathcal{Z}_{\text{trn}}, \mathcal{Z}_{\text{aug}}, \mathcal{Z}_{\text{test}}) = 1$ if and only if $u_1 = u_2 = 0$ when $\delta = 1$. First, we represent the embedding vector (which is used to be a matrix) $\mathbf{z} \in \mathbb{R}^4$ before the total distance normalization as

$$\mathbf{z} = (0, u_1, 2, u_2 + 2).$$

Let $\bar{z} = (u_1 + u_2 + 4)/4$ be the center, and $\bar{\mathbf{z}} \in \mathbb{R}^4$ be a vector where each element is $\bar{z}$. Then, $\mathbf{z}$ is transformed into $\tilde{\mathbf{z}}$ as a result of the normalization:

$$
\begin{aligned}
\tilde{\mathbf{z}} &= \frac{2}{\sqrt{\bar{z}^2 + (u_1 - \bar{z})^2 + (2 - \bar{z})^2 + (u_2 + 2 - \bar{z})^2}}(\mathbf{z} - \bar{\mathbf{z}}) \\
&= \frac{2}{\sqrt{4\bar{z}^2 - 2(u_1 + u_2 + 4)\bar{z} + u_1^2 + u_2^2 + 4u_2 + 8}}(\mathbf{z} - \bar{\mathbf{z}}) \\
&= \frac{2}{\sqrt{u_1^2 + u_2^2 + 4u_2 + 8 - 4\bar{z}^2}}(\mathbf{z} - \bar{\mathbf{z}}) \\
&= \frac{4}{\sqrt{3u_1^2 + 3u_2^2 - 8u_1 + 8u_2 - 2u_1u_2 + 16}}(\mathbf{z} - \bar{\mathbf{z}}).
\end{aligned}
$$

The validation loss $\mathcal{L}_{\text{val}}$ is computed on $\tilde{\mathbf{z}}$ as follows:

$$\mathcal{L}_{\text{val}}(\mathcal{Z}_{\text{trn}}, \mathcal{Z}_{\text{aug}}, \mathcal{Z}_{\text{test}}) = \frac{|u_1| + |u_1 - 2| + |u_2| + |u_2 + 2|}{\sqrt{3u_1^2 + 3u_2^2 - 8u_1 + 8u_2 - 2u_1u_2 + 16}}.$$

Then, we can consider four cases based on whether $u_1 \geq 0$ and $u_2 \geq 0$. To show that the inequality $\mathcal{L}_{\text{val}}(\mathcal{Z}_{\text{trn}}, \mathcal{Z}_{\text{aug}}, \mathcal{Z}_{\text{test}}) \geq 1$ holds, we represent $\mathcal{L}_{\text{val}}^2(\mathcal{Z}_{\text{trn}}, \mathcal{Z}_{\text{aug}}, \mathcal{Z}_{\text{test}}) - 1$ as follows:

$$\mathcal{L}_{\text{val}}^2(\mathcal{Z}_{\text{trn}}, \mathcal{Z}_{\text{aug}}, \mathcal{Z}_{\text{test}}) - 1 = \begin{cases} -3u_1^2 + u_2^2 + 8u_1 + 8u_2 + 2u_1u_2 & \text{if } u_1 \geq 0 \text{ and } u_2 \geq 0 \\ -3u_1^2 - 3u_2^2 + 8u_1 - 8u_2 + 2u_1u_2 & \text{if } u_1 \geq 0 \text{ and } u_2 \leq 0 \\ u_1^2 + u_2^2 - 8u_1 + 8u_2 - 6u_1u_2 & \text{if } u_1 \leq 0 \text{ and } u_2 \geq 0 \\ u_1^2 - 3u_2^2 - 8u_1 - 8u_2 + 2u_1u_2 & \text{if } u_1 \leq 0 \text{ and } u_2 \leq 0 \end{cases}$$

It is straightforward to see that $\mathcal{L}_{\text{val}}^2(\mathcal{Z}_{\text{trn}}, \mathcal{Z}_{\text{aug}}, \mathcal{Z}_{\text{test}}) - 1 \geq 0$ in all four cases if $-1 \leq u_1 \leq 1$ and $-1 \leq u_2 \leq 1$, and the equality holds if and only if $u_1 = u_2 = 0$. Since $\mathcal{L}_{\text{val}}(\mathcal{Z}_{\text{trn}}, \mathcal{Z}_{\text{aug}}, \mathcal{Z}_{\text{test}}) \geq 0$ by its definition, we prove the lemma for the scalar case. The extension to multi-dimensional cases is trivial, since all operations in $\mathcal{L}_{\text{val}}$ can be generalized to arbitrary dimensions. $\square$

## C DETAILED INFORMATION OF CUTDIFF

### C.1 VISUALIZATION WITH DIFFERENT HYPERPARAMETERS

Fig. 8 shows the images generated from CutPaste, CutOut, and CutDiff, respectively, with different hyperparameter choices. CutDiff is similar to CutOut, except that it creates a smooth circular patch which supports gradient-based updates of its hyperparameters. It is clear from the figure that the two other augmentations, CutPaste and CutOut, are not differentiable as they replace the original pixels with new ones (either black for CutOut or a copied patch for CutPaste).

### C.2 DETAILED EXPLANATION ON THE CUTDIFF ALGORITHM

We provide more details of CutDiff in Alg. 1 and discuss what makes it differentiable unlike existing augmentation functions like CutOut. Recall that image augmentation is a function of pixel positions, rather than pixel values; although the output is determined also by the input image, the pixel values do not change how the augmentation performs. The main idea of CutDiff is to introduce a grid $\mathbf{G}$ of pixel locations as in Line 3 of Alg. 1 and design a differentiable function that takes $\mathbf{G}$ as an input and determines how to augment each pixel location based on the augmentation hyperparameters $\mathbf{a}$.

The process contains three steps. First, we sample the center position $\mu$ as in Line 4 of Alg. 1, which is a constant with respect to $\mathbf{a}$. Then, for each position $(i, j)$, we determine the amount of change to

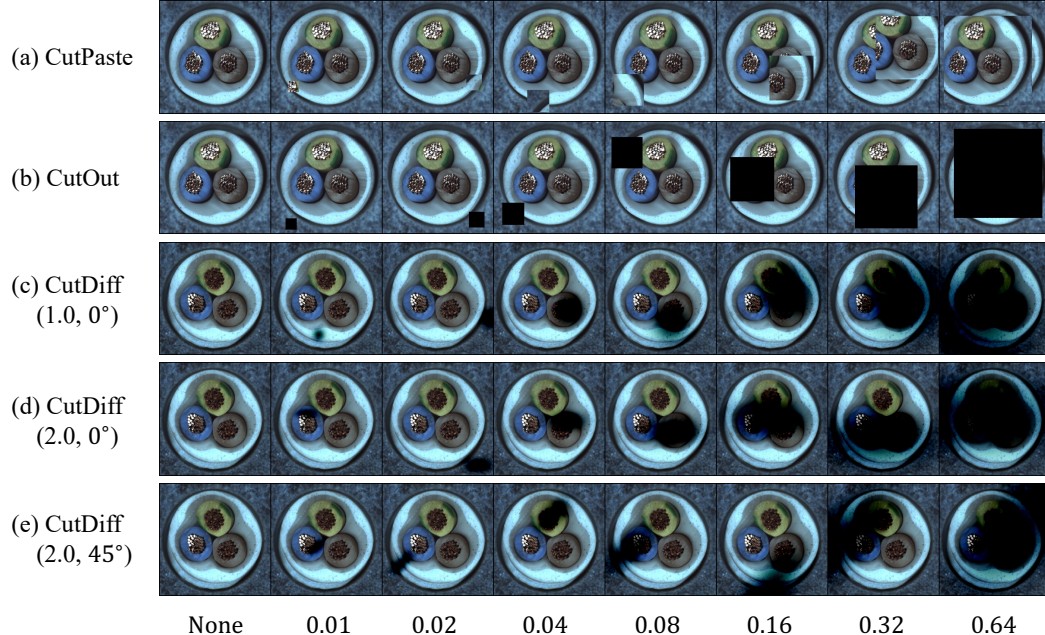

|  | None | 0.01 | 0.02 | 0.04 | 0.08 | 0.16 | 0.32 | 0.64 |

Figure 8: Comparison between three local augmentation functions: (a) CutPaste, (b) CutOut, and (c - e) CutDiff with different hyperparameter choices. Each column represents a different patch size in $[0.01, 0.64]$, and the two numbers in (c - e) represent the aspect ratio and the rotated angle of a patch, respectively. CutDiff creates a smooth differentiable boundary unlike the other two functions.

make with augmentation based on the distance from $\mu$ scaled by $\mathbf{a}$ (in Line 5). In other words, the amount of change is small if $(i, j)$ is far from $\mu$, where the exact value is affected also by $\mathbf{a}$. Lastly, we replace the assignment (or replacement) operation in CutOut (and CutPaste) with the subtract operation (in Line 6). The $\min$ and $\max$ operations are used to ensure that the output pixels are in $[0, 1]$. Thanks to the subtraction, the information of the original pixels, not only the pixel locations, is passed to the output image, allowing the gradient to flow to both the given image $\mathbf{x}$ and $\mathbf{a}$.

It is noteworthy that we can also model the patch location $\mu$ as a hyperparameter in $\mathbf{a}$. The problem is that it makes a too strong assumption that a defect is located similarly in all images, which may not be true in many cases. The randomness of $\mu$ does not harm the gradient-based optimization of ST-SSAD, because the validation loss $\mathcal{L}_{\text{val}}$ is computed on a set of embeddings at once, rather than on each individual sample, which may cause the instability of optimization process.

## D  DETAILED EXPERIMENTAL SETUP

### D.1  IMPLEMENTATION DETAILS

ST-SSAD has two techniques in its implementation which are not introduced in the main paper due to the lack of space. The first is a *warm start,* which means that we train the detector $f_\theta$ for a fixed number of epochs before starting the alternating updates of $\theta$ and the augmentation hyperparameters $\mathbf{a}$. The warm start is required since the gradient-based update of $\mathbf{a}$ is ineffective if the detector $f_\theta$ does not perform well on the current $\mathbf{a}$; in such a case, the validation loss $\mathcal{L}_{\text{val}}$ can have an arbitrary value which is not related to the true alignment of $\mathbf{a}$. The number of training epochs for warm start is chosen to sufficiently minimize the training loss $\mathcal{L}_{\text{trn}}$ for the initial $\mathbf{a}$.

The second technique is to update $\theta$ multiple times for each update of $\mathbf{a}$. This is based on the same motivation as in the first technique; we need to ensure the reasonable performance of $f_\theta$ during the iterative updates. The default choice is only one update of $\theta$, but when the training loss $\mathcal{L}_{\text{trn}}$ does not decrease enough at each iteration, one needs to consider increasing the number of updates.

Table 3: Ablation studies of our ST-SSAD on 8 different tasks: (left) varying sizes of augmentation data, (middle) disjoint validation and test data, and (right) varying ratios of anomalies in the training data. ST-SSAD shows consistent results in terms of both average accuracy and average rank.

| Object | Anomaly Type | 64 | 128 | 256 (Original) | 512 | Shared (Original) | Disjoint | 0% (Original) | 1% | 2% |
|---|---|---|---|---|---|---|---|---|---|---|
| Cable | Bent wire | 0.521 (3) | 0.596 (1) | 0.490 (4) | 0.575 (2) | 0.490 (2) | 0.577 (1) | 0.490 (3) | 0.711 (1) | 0.581 (2) |
| Cable | Cable swap | 0.573 (2) | 0.579 (1) | 0.532 (4) | 0.573 (2) | 0.532 (2) | 0.648 (1) | 0.532 (3) | 0.642 (1) | 0.626 (2) |
| Cable | Combined | 0.913 (2) | 0.878 (4) | 0.925 (1) | 0.887 (3) | 0.925 (1) | 0.879 (2) | 0.925 (1) | 0.880 (2) | 0.868 (3) |
| Cable | Cut inner insulation | 0.717 (1) | 0.631 (4) | 0.667 (2) | 0.641 (3) | 0.667 (2) | 0.723 (1) | 0.667 (2) | 0.753 (1) | 0.585 (3) |
| Cable | Cut outer insulation | 0.457 (4) | 0.525 (1) | 0.516 (3) | 0.525 (1) | 0.516 (2) | 0.639 (1) | 0.516 (3) | 0.603 (2) | 0.607 (1) |
| Cable | Missing cable | 0.909 (4) | 0.973 (2) | 0.998 (1) | 0.967 (3) | 0.998 (1) | 0.946 (2) | 0.998 (1) | 0.905 (2) | 0.789 (3) |
| Cable | Missing wire | 0.507 (4) | 0.706 (3) | 0.863 (1) | 0.723 (2) | 0.863 (1) | 0.736 (2) | 0.863 (1) | 0.686 (3) | 0.826 (2) |
| Cable | Poke insulation | 0.722 (1) | 0.683 (2) | 0.630 (4) | 0.631 (3) | 0.630 (2) | 0.681 (1) | 0.630 (3) | 0.649 (2) | 0.674 (1) |
| | Average AUC | 0.665 | 0.696 | 0.703 | 0.690 | 0.703 | 0.729 | 0.703 | 0.729 | 0.695 |
| | Average Rank | 2.6 | 2.3 | 2.5 | 2.4 | 1.6 | 1.4 | 2.1 | 1.8 | 2.1 |

## D.2 HYPERPARAMETER CHOICES

We choose the model and training hyperparameters of ST-SSAD mostly based on a previous work (Li et al., 2021), including the detector network $f_\theta$ and the score function $s$, and use them across all tasks in our experiments. On the other hand, we tune some of the hyperparameters that affect the effectiveness of training and need to be controlled based on the property of each dataset:

- Batch size: 32 (in MVTecAD) and 256 (in SVHN)
- Number of epochs for warm start: 20 (in MVTecAD) and 40 (in SVHN)
- Number of updates for detector parameters $\theta$: 1 (in MVTec) and 5 (in SVHN)
- Number of maximum iterations: 500 (in MVTecAD) and 100 (in SVHN)

Batch size is set to a small number in MVTecAD, since the images in the dataset have high resolution $256 \times 256$, causing high memory cost, while the number of samples is small in both training and test data. The number of epochs for warm start and the number of updates for $\theta$ are set large in SVHN, since it has more diverse images than in MVTecAD and requires more updates of $\theta$ to decrease the training loss $\mathcal{L}_{\mathrm{trn}}$ sufficiently. The number of maximum iterations is set differently so that the total number of updates for $\theta$ is the same in both datasets. Please find more details on our implementation from our anonymized code repository: https://anonymous.4open.science/r/ST-SSAD.

It is noteworthy that the choice of those hyperparameters is done by observing the training process, especially the speed that the training loss $\mathcal{L}_{\mathrm{trn}}$ is minimized before and during the iterations, rather than the actual performance of ST-SSAD, which is not accessible in unsupervised AD tasks.

## D.3 COMPUTATIONAL ENVIRONMENT

All our experiments were done in a shared computing center containing NVIDIA Tesla V100 GPUs, Intel Xeon Gold 6248 CPUs, and 512GB DDR4-2933 memory.

# E MORE EXPERIMENTAL RESULTS

We give in Table 3 the results of three additional experiments on ST-SSAD: 1) varying augmentation data sizes, 2) validation-test split, and 3) contaminated training data. All experiments were done in the MVTec dataset, specifically the Cable object and the eight anomaly types associated with it.

## E.1 VARYING AUGMENTATION DATA SIZES

Our validation loss $\mathcal{L}_{\mathrm{val}}$ is designed to be robust to $|\mathcal{D}_{\mathrm{aug}}|/|\mathcal{D}_{\mathrm{trn}}|$, since the ratio of true anomalies in test data is unknown at training time. Consequently, we simply set $|\mathcal{D}_{\mathrm{aug}}| = |\mathcal{D}_{\mathrm{trn}}|$ in all experiments, such that we perform one augmentation per training example. Specifically, instead of using all training data in every computation of $\mathcal{L}_{\mathrm{val}}$, we randomly sample 256 training samples (and 256 augmentation samples) at each computation for efficiency. The number 256 is chosen large enough to estimate the true distribution of training data.

In this experiment, we vary the number of augmentation samples that are used in the computation of $\mathcal{L}_{\mathrm{val}}$ in $\{64, 128, 256, 512\}$, where 256 is the choice in the original experiments. The first four

columns in Table 3 show the result. The performance is not sensitive to the size of augmented data, and we obtain similar results across eight different tasks.

### E.2 VALIDATION-TEST SPLIT

Although we focus on transductive anomaly detection tasks, where unlabeled test data are given at training time, we perform an additional experiment where we split the original test data into two disjoint sets of validation and (new) test data, with the size ratio 1:1, and use only the validation data in the computation of our alignment loss. This is to create an evaluation setting where the evaluation data (i.e., new test data) are completely separated from the data observed in training time.

The middle two columns in Table 3 show the result. We find a negligible difference between the two settings, demonstrating that our approach succeeds with the separated test data.

### E.3 CONTAMINATED TRAINING DATA

In the main experiments, we assume a problem setting where training data consist of only normal samples, as done in many previous works on self-supervised anomaly detection. However, one may consider this setting unrealistic, since acquiring pure clean data is often hard in real-world scenarios. We thus perform experiments with contaminated training data by including a small number of anomalies in it. We increase the ratio of contamination up to 2%, since the number of anomalies in the MVTec dataset is not large enough to increase more.

The right three columns in Table 3 show the result. The overall performance is similar in the three settings, showing that ST-SSAD is robust to the small fraction of anomalies in training data.

