# OpenReview forum: "Self-Tuning Self-Supervised Anomaly Detection"
_ICLR.cc/2024/Conference — Submitted to ICLR 2024_

### Official Review · Reviewer_iRPB · 2023-10-27

**Soundness:** 1 poor
**Presentation:** 3 good
**Contribution:** 3 good
**Rating:** 6
**Confidence:** 4

**Summary:**

This paper mainly studies an adaptive method to search the optimal data augmentation function for self-supervised anomaly detection task. It formulates the learning of potential abnormal distribution (i.e. true anomaly-generating mechanism) as a second-order optimization problem. To this end, two aspects are mainly studied: (1) a new validation loss for differentiable distribution matching of augmented training data and the unlabeled testing anomalies; (2) some differentiable augmentation functions for optimizing their learnable control factors and detector parameters alternatively. Experiments on both semantic and non-semantic anomalies demonstrates the effectiveness.

**Strengths:**

- This paper is well-motivated and clearly presented. Researchers could gain a lot of insight into this paradigm and the potential impact on discriminative learning with synthetic anomaly is significant. Besides, the authors provide theoretical evidence and clear illustrations for better understanding, including their idea, method and some demonstrative examples.

- The optimal augmentation function seems to alleviate the limitation and difficulty of artificially synthesized pseudo anomalies, due to the better consistency with real anomalies.

- The parameters are optimized during end-to-end training, which is feasible and ideal.

**Weaknesses:**

- My major concern of this paper lies in the strong assumption, here are a couple of aspects:

1.	The authors evaluate MV Tech AD with the data being collected according to given abnormal patterns. I wonder if the proposed framework can simultaneously deal with multiple different augmentations, since this case is more general in real-world anomaly detection scenarios and is more likely to avoid overfitting to the optimal **a**. It will be helpful to show the results with **UN**split original classes in MV Tech AD.

2.	The assumption that the validation and test distributions are consistent may be too strong for the setting of anomaly detection task (*e.g.* new anomaly pattern or even mixing anomalies makes **a** not optimal). It may not be reasonable to omit test images drawn from unseen distributions in real scenarios.

- The author believes that the heuristic function $S$ will be high (higher variance of anomaly scores) *only if* augmentation parameters are initialized better (more separable distributions). However, this may not always hold, as the optimization of a mismatched *strong/hard* augmentation (as long as the decision boundary divides testing distribution into any two parts) may also lead to high variance of anomaly scores. I wonder if there are any theoretical or intuitive explanations about this issue.

- The authors use CutDiff and Rot for non-semantic and semantic shift detection respectively. But I would like to know could ST-SSAD learn zero-angle Rot for non-semantic defects and zero-size CutDiff for semantic shifts in end-to-end optimization when using both augmentations together.

- The term ``Ratio invariance’’ may be imprecise. With the augmentation quantity changed, $L_{val}$ could be changed together because the included $\frac{\sqrt{N}}{{\Vert {Z}^{c}\Vert}_{F}}$ is different (I notice that this have no negative impact on optimization target).

- There are some writing issues, *e.g.* superscript $^{n}$ should be $^{a}$ in **Lemma 2**.

**Questions:**

Please refer to the "weaknesses" part for details.

---

> ### Author Response · Authors · 2023-11-19
> **Response to Reviewer iRPB**
>
> We thank the reviewer for the constructive and detailed feedback. We provide our answers to the weaknesses and questions pointed out by the reviewer. Some of our answers point to the common responses.
>
> > **W1.** The authors evaluate MV Tech AD with the data being collected according to given abnormal patterns. I wonder if the proposed framework can simultaneously deal with multiple different augmentations, since this case is more general in real-world anomaly detection scenarios and is more likely to avoid overfitting to the optimal a. It will be helpful to show the results with UNsplit original classes in MV Tech AD.
>
> Our current approach can be subpar on test data that contains anomalies of multiple types, for which tuning the discrete choice of augmentation functions becomes a new challenge.  Our work presents a framework to tune the *continuous* hyperparameters of an augmentation function, provided with the discrete choice. Please refer to **C2** of our common responses for possible hyperparameter optimization (HPO) approaches we laid out as future work.
>
> > **W2.** The assumption that the validation and test distributions are consistent may be too strong for the setting of anomaly detection task (e.g. new anomaly pattern or even mixing anomalies makes a not optimal). It may not be reasonable to omit test images drawn from unseen distributions in real scenarios.
>
> We believe that transductive learning is common in real-world anomaly detection tasks. Please  refer to **C1** of our common responses for an extended discussion on transductive vs. inductive learning in the context of anomaly detection. Nevertheless, we also conducted additional experiments dividing unlabeled test data into validation and (new) test data, and then using only the validation data for computing the alignment loss. Please refer to **C3 #2** of our common responses for the results.
>
> > **W3.** The author believes that the heuristic function $S$ will be high (higher variance of anomaly scores) only if augmentation parameters are initialized better (more separable distributions). However, this may not always hold, as the optimization of a mismatched strong/hard augmentation (as long as the decision boundary divides testing distribution into any two parts) may also lead to high variance of anomaly scores. I wonder if there are any theoretical or intuitive explanations about this issue.
>
> We remark that this heuristic is employed only after optimizing for the alignment first. In other words, we use it only to choose from among models that are already optimized to align with the test data as well as possible. Since a decision boundary dividing the test data would not yield a low validation loss in the first place, the heuristic is not likely to be exposed to such scenarios in practice.
>
> > **W4.** The authors use CutDiff and Rot for non-semantic and semantic shift detection respectively. But I would like to know could ST-SSAD learn zero-angle Rot for non-semantic defects and zero-size CutDiff for semantic shifts in end-to-end optimization when using both augmentations together.
>
> As discussed in detail in **C2** of our common responses, our current model is designed to tune the *continuous* hyperparameters of a given augmentation function. Given a set of augmentation functions, making a discrete choice between the different functions or optimizing a mixture of functions together poses a new challenge. For example, with both CutDiff and Rotate included, the ratio of augmented samples to generate with the two functions becomes another hyperparameter. Being the first solution to the automatic model selection for SSAD, we expect our work will trigger several follow-up works.
>
> > **W5.** The term “Ratio invariance” may be imprecise. With the augmentation quantity changed,  $L_\mathrm{val}$  could be changed together because the included $\sqrt{N} / \| Z^C \|_F$ is different (I notice that this have no negative impact on optimization target).
>
> That is correct. The total distance normalization is affected by the size of augmented data, and thus our loss function is not exactly invariant to it. However, the effect is negligible, since that mainly changes the scale of embeddings, rather than their distributions. For preciseness, we have changed the term into "ratio robustness" in the revised manuscript.
>
> > **W6.** There are some writing issues, e.g. superscript $^n$ should be $^a$ in Lemma 2.
>
> Thank you for finding out the typo. We have updated the revised manuscript accordingly.

---

### Official Review · Reviewer_ht7A · 2023-10-30

**Soundness:** 3 good
**Presentation:** 3 good
**Contribution:** 3 good
**Rating:** 6
**Confidence:** 3

**Summary:**

The authors propose learnable augmentations for transductive anomaly
detection.  In the transductive scenario, the test set, including
anomalies to be detected, are available for training.  They called
their method Self-Tuning Self-Supervsied Anomaly Detection (ST-SSAD).
The augmentations help simulate anomalies.  Using the proposed
validation loss, ST-SSAD tries to make the original training instances
together with their augmentations, similar to the test set.  The
validation loss is based on the distance, in the representation space,
between each test instance and the mean of training instances, as well
as the mean of augmented instances.  Instances in representation space
are transformed to have unit total pairwise squared distance.  Binary
cross entropy loss is used as the training loss.  For learnable
augmentations, they proposed differentiable CutDiff (local
augmentation) and rotation (global augmentation).

ST-SSAD was compared with multiple baseline algorithms on two datasets
with different types of anomalies. Empirical results indicate ST-SSAD
generally outperforms the baselines.  Ablation studies indicate the
proposed components of ST-SSAD contribute to higher performance.

**Strengths:**

Allowing augmentations to be learnable/differentiable is interesting.
Examples were presented to show augmentations can match anomalies.
Empirical results indicate ST-SSAD generally outperforms the
baselines.  The paper is generally well written.

**Weaknesses:**

During evaluation, each anomaly type is separated.  The proposed
ST-SSAD seems to assume only one anomaly type exists in the test set.
That is, the user might need to use ST-SSAD for each anomaly type.
How to handle multiple anomaly types in the same test set is not
clear.

Details are in questions below.

**Questions:**

1.  In the experiments, anomaly types are separately evaluated.  That
seems to mean that the anomaly type is known, and the augmentation
parameters are learned to match the anomaly type.  This seems to be
assumed in Equation 3 because the second term calculates the mean of
the augmented instances.  That is, Eq. 3 seems to be finding an
augmentation and its parameters to match the anomaly type.  If that is
correct, how can multiple types of anomalies in the same test set be
handled?

2.  While the augmentation parameters are learnable, the augmentation
types such as cut and rotation need to be specified.  Also, to be
detected, seemingly an anomaly type in the test set needs to match one
of the augmentation types.  Could the matching be relaxed?  That is,
the user potentially does not need to know what the anomaly types are.

3.  What is the size of $D_{aug}$?

4.  Eq 2, equation on the right for $z_i^c$: should $z$ be $z_i$ and
the summation is over another index such as $j$ to not confuse with
$i$?

5.  Eq 3: mean(.) seems to be similar to $1/N\sum_{i=1}^{N}z_{i}$ in
Eq 2.  If so, using mean(.) in Eq 2 would make the presentation
consistent.  If not, what is the difference?

---

> ### Author Response · Authors · 2023-11-19
> **Response to Reviewer ht7A**
>
> We thank the reviewer for the constructive and detailed feedback. We provide our answers to the weaknesses and questions pointed out by the reviewer. Some of our answers point to the common responses.
>
> > **W1.** During evaluation, each anomaly type is separated. The proposed ST-SSAD seems to assume only one anomaly type exists in the test set. That is, the user might need to use ST-SSAD for each anomaly type. How to handle multiple anomaly types in the same test set is not clear.
>
> > **Q1.** In the experiments, anomaly types are separately evaluated. That seems to mean that the anomaly type is known, and the augmentation parameters are learned to match the anomaly type. This seems to be assumed in Equation 3 because the second term calculates the mean of the augmented instances. That is, Eq. 3 seems to be finding an augmentation and its parameters to match the anomaly type. If that is correct, how can multiple types of anomalies in the same test set be handled?
>
> As the reviewer accurately pointed out, our approach does not currently address test data that contain anomalies of multiple types, for which tuning the discrete choice of augmentation functions becomes a new challenge. Our work presents a framework to tune the *continuous* hyperparameters of an augmentation function, provided with the discrete choice. We refer to **C2** of our common responses for possible hyperparameter optimization (HPO) approaches we laid out as future work toward tackling a mixture of anomalies in test data.
>
> > **Q2.** While the augmentation parameters are learnable, the augmentation types such as cut and rotation need to be specified. Also, to be detected, seemingly an anomaly type in the test set needs to match one of the augmentation types. Could the matching be relaxed? That is, the user potentially does not need to know what the anomaly types are.
>
> As we show in Section 4.4, our approach succeeds if the augmentation function is aligned with the true anomalies at the functional level. As discussed in **C2** of our common responses, the *discrete* choice of the augmentation from a *set* of possible functional forms can be addressed via black-box hyperparameter optimization (HPO). Our work presents a framework to tune the *continuous* hyperparameters provided with the discrete choice. We expect several follow-up works, not only proposing new differentiable augmentation functions but also more elaborate HPO techniques to select from this set, as laid out in **C2.**
>
> > **Q3.** What is the size of $D_{aug}$?
>
> We set the size of $D_{aug}$ to be the same as that of $D_{trn}$ in all experiments. This was to avoid introducing an additional hyperparameter. To show robustness, we conducted additional experiments varying the ratio of augmented data over the size of training data. Please refer to **C3 #1** of our common responses for the results.
>
> > **Q4.** Eq 2, equation on the right for $z_i^c$: should $z$ be $z_i$ and the summation is over another index such as $j$ to not confuse with $i$?
>
> That’s correct. We have changed Eq. (2) accordingly in the revised manuscript. Thank you for pointing it out.
>
> > **Q5.** Eq 3: mean(.) seems to be similar to $1/N \sum_{i=1}^N z_i$ in Eq 2. If so, using mean(.) in Eq 2 would make the presentation consistent. If not, what is the difference?
>
> Both expressions are almost the same: mean(.) in Eq. (3) and $1/N \sum_{i=1}^N z_i$ in Eq. (2). The only difference is that mean(.) in Eq. (3) is an operator on a set of vectors, while in Eq. (2) we computed the mean of rows in a matrix. It would make them consistent if we treat both as sets or both as matrices, but we believe the current way is more intuitive and easier to understand.

---

> > ### Comment · Reviewer_ht7A · 2023-11-22
> > **comments on authors' response**
> >
> > Thanks for the response.  C2 has some preliminary ideas, and results from them would strengthen the paper.

---

> > > ### Author Response · Authors · 2023-11-22
> > > **Reply to Reviewer ht7A**
> > >
> > > Thank you for your valuable feedback. We hope our work serves as a meaningful foundation for future works that further concretize and develop the ideas presented in C2.

---

### Official Review · Reviewer_gndU · 2023-10-31

**Soundness:** 2 fair
**Presentation:** 3 good
**Contribution:** 2 fair
**Rating:** 5
**Confidence:** 4

**Summary:**

This paper introduces ST-SSAD, a novel approach for self-supervised anomaly detection (SSAD). It addresses the challenge of selecting proper data augmentation functions to generate pseudo-anomalies that are close to real anomalies. ST-SSAD offers two main contributions: an unsupervised validation loss using an unlabeled test dataset for tuning augmentation and differentiable augmentation functions for end-to-end hyperparameter tuning. Experimental results on two testbeds demonstrate performance improvements through the systematic augmentation tuning.

**Strengths:**

- The paper addresses the challenge of end-to-end augmentation tuning in SSAD.
- The idea of employing differentiable augmentation, such as CutDiff, is interesting and demonstrates potential applicability to other domains beyond anomaly detection.
- The paper is well organized and clearly written overall.

**Weaknesses:**

- A major concern is that ST-SSAD replies on the entire test dataset during training and tuning. While the authors mention transductive learning, this approach is not quite realistic, particularly in the context of anomaly detection. The tuning result will be overly sensitive to the specific anomaly types in the test data and may not generalize well. The paper lacks clarification, experimental results, or in-depth discussion on this issue, which significantly limits the applicability and advantages of the proposed method.

- The mean distance loss is proposed for ratio invariance with theoretical properties, but no experimental result validating this invariance is provided.

- The method still requires prior knowledge about anomalies and heavily depends on it. For example, the augmentation functions of either local (CutDiff) or global (rotation) augmentations are considered and therefore, it works well only when anomalies closely resemble specific shapes that these functions can reflect. The method will also fail in the case where rotated samples are considered normal.

- The authors state that 'we focus on the performance of each anomaly type rather than overall accuracy.' However, in real-world scenarios, it is common to encounter various types of anomalies. Therefore, it will be more crucial to investigate such practical scenarios.

- It was mentioned that "there are no direct competitors on end-to-end augmentation hyperparameter tuning..."; however, it is essential to include performance comparison with the latest models that clearly distinguish train and test data. The results in Tables 1 and 2 appear to be more like an ablation study, so it is difficult to assess whether the proposed method truly outperforms the latest models in a meaningful way.

**Questions:**

- Can you provide results using a validation set that is disjoint from the test set?

- I guess the proposed method may be quite sensitive to the proportion of abnormal samples in the test set. Can you provide experimental results or a discussion addressing this issue? And please provide the ratio between normal and abnormal samples in the presented results of the current manuscript.

- Is it possible for the model to learn effectively in a scenario where abnormal samples are inherently present in the training set but remain unlabeled?

---

> ### Author Response · Authors · 2023-11-19
> **Response to Reviewer gndU**
>
> We thank the reviewer for the constructive and detailed feedback. We provide our answers to the weaknesses and questions pointed out by the reviewer. Some of our answers point to the common responses.
>
> > **W1.** A major concern is that ST-SSAD replies on the entire test dataset during training and tuning. While the authors mention transductive learning, this approach is not quite realistic, particularly in the context of anomaly detection. The tuning result will be overly sensitive to the specific anomaly types in the test data and may not generalize well. The paper lacks clarification, experimental results, or in-depth discussion on this issue, which significantly limits the applicability and advantages of the proposed method.
>
> We believe that transductive learning is common in real-world anomaly detection tasks, whereas generalization is a matter for inductive learning. Please refer to **C1** of our common responses.
>
> > **W2.** The mean distance loss is proposed for ratio invariance with theoretical properties, but no experimental result validating this invariance is provided.
>
> We have conducted additional experiments varying the ratio of augmented data over the size of training data. Please refer to **C3 #1** of our common responses for the result.
>
> > **W3.** The method still requires prior knowledge about anomalies and heavily depends on it. For example, the augmentation functions of either local (CutDiff) or global (rotation) augmentations are considered and therefore, it works well only when anomalies closely resemble specific shapes that these functions can reflect. The method will also fail in the case where rotated samples are considered normal.
>
> As we discussed in Section 4.4, our approach succeeds if the augmentation function is aligned with the true anomalies at the functional level. Nevertheless, as the first systematic approach to the automatic model selection for self-supervised anomaly detection, we believe our work can lead to numerous future works that extend our approach to new differentiable augmentations that capture other types of real-world anomalies. Please refer to **C2** of our common responses for more discussion on the problem setup and suggested directions of future works.
>
> > **W4.** The authors state that 'we focus on the performance of each anomaly type rather than overall accuracy.' However, in real-world scenarios, it is common to encounter various types of anomalies. Therefore, it will be more crucial to investigate such practical scenarios.
>
> Our current approach does not address test data that contain anomalies of multiple types, for which tuning the discrete choice of augmentation functions becomes a new challenge. Our work presents a framework to tune the *continuous* hyperparameters of an augmentation function, provided with the discrete choice. As we answered to W3, please refer to **C2** of our common responses for possible hyperparameter optimization (HPO) approaches we laid out as future work toward tackling a mixture of anomalies in test data.
>
> > **W5.** It was mentioned that "there are no direct competitors on end-to-end augmentation hyperparameter tuning..."; however, it is essential to include performance comparison with the latest models that clearly distinguish train and test data. The results in Tables 1 and 2 appear to be more like an ablation study, so it is difficult to assess whether the proposed method truly outperforms the latest models in a meaningful way.
>
> Please note that it is difficult to conduct a fair comparison between our approach and other works on anomaly detection, since the extent of “fair” hyperparameter tuning is hard to define between different lines of works. A vast majority of work on unsupervised anomaly detection does not address the unsupervised hyperparameter (HP) tuning problem at all.
>
> Our results show significant performance boost over two most prominent unsupervised approaches; DeepSVDD (one-class) and DeepAE (reconstruction-based). On the self-supervised side, we show that tuning augmentation significantly outperforms arbitrarily choosing the augmentation, including the CutOut and CutPaste approaches (CO and CP in Table 1) proposed by Devries & Taylor (2017) and Li et al. (2021), respectively.  Since we used the detector network and the score function used in (Li et al., 2021) throughout all experiments, we can consider the RS-CP and RD-CP baselines in Table 1 as "untuned" versions of the CutPaste approach (Li et al., 2021).

---

> ### Author Response · Authors · 2023-11-19
> **Response to Reviewer gndU (continued)**
>
> > **Q1.** Can you provide results using a validation set that is disjoint from the test set?
>
> Yes, we have conducted additional experiments dividing unlabeled test data into validation and (new) test data, and then using only the validation data for computing the alignment loss. Please refer to **C3. #2** of our common responses for the results.
>
> > **Q2.** I guess the proposed method may be quite sensitive to the proportion of abnormal samples in the test set. Can you provide experimental results or a discussion addressing this issue? And please provide the ratio between normal and abnormal samples in the presented results of the current manuscript.
>
> Our datasets already vary in the proportion of anomalies they contain in the test data. The following table shows the percentage of anomalies across different tasks in the MVTec dataset, where the minimum and maximum are 0.147 and 0.404, respectively. Similarly in the SVHN dataset, the fraction is between 0.279 and 0.722, since the different classes contain different numbers of examples. Thus, we believe that our approach is not sensitive to the ratio of anomalies in test data, as long as the anomalies are separable from the normal data and thus the alignment can be measured by our validation loss.
>
> obj_type | ano_type | fraction
> --- | --- | ---
> cable | bent_wire | 0.183
> cable | cable_swap | 0.171
> cable | combined | 0.159
> cable | cut_inner_insulation | 0.194
> cable | cut_outer_insulation | 0.147
> cable | missing_cable | 0.171
> cable | missing_wire | 0.147
> cable | poke_insulation | 0.147
> carpet | color | 0.404
> carpet | cut | 0.378
> carpet | hole | 0.378
> carpet | metal_contamination | 0.378
> carpet | thread | 0.404
> grid | bent | 0.364
> grid | broken | 0.364
> grid | glue | 0.344
> grid | metal_contamination | 0.344
> grid | thread | 0.344
> tile | crack | 0.340
> tile | glue_strip | 0.353
> tile | gray_stroke | 0.327
> tile | oil | 0.353
> tile | rough | 0.312
>
> > **Q3.** Is it possible for the model to learn effectively in a scenario where abnormal samples are inherently present in the training set but remain unlabeled?
>
> Yes, this refers to the setting where training data is not “clean” (i.e. anomaly-free) but is contaminated with some anomalies. We have conducted additional experiments by varying the degree of contamination in the training data. Please refer to **C3 #3** of our common responses for the results.

---

### Author Response · Authors · 2023-11-19
**Common Responses**

We thank the reviewers for the detailed and constructive reviews. Before responding to the individual reviews, we give common responses on important aspects of our paper.

## C1. Transductive Anomaly Detection

In machine learning, transductive learning and inductive learning refer to two different scenarios of training and evaluating a model. In transductive learning, we train a model given unlabeled test data that we aim to predict; while in inductive learning, we train a model given only training data. Generalization to unseen data is not an important issue in transductive learning, since when new data are given at test time, one can retrain the model on the new data.

Our work solves anomaly detection in the transductive setting. Given training and unlabeled test data, we train a detector to accurately predict the unknown labels of test data. Many real-world tasks for anomaly detection are transductive: for example, 1) given the annual transaction records from a firm, find abnormal ones, 2) given a database of patient records, find rare patients, 3) given millions of human mobility trajectories, find unusual trajectories, and so on. If a new dataset was given in the future, we would repeat our approach on that data again, rather than employing a historically trained detector inductively.

As such, there is no validation set in transductive anomaly detection since we don’t have any labels to use for validation. Thus, we do the hyperparameter tuning *specific* to the input test data, rather than general unseen data. In fact, it is a core stance of our work to tune augmentation to the test data at hand transductively, rather than choosing augmentation arbitrarily by “imagining” what unseen anomalies would look like.

Nevertheless, we have conducted additional experiments where we separate the original test data into disjoint sets of validation and (new) test data. We use only the validation data for computing our alignment loss to make sure that the test data do not participate in any part of the training process, although it is different from typical transductive learning. Note that both the validation and test data are unlabeled in our case. The result is given in **C3 #2** of our common responses, and it shows a negligible difference from the original result in the paper.

## C2. Multiple Anomaly Types

There are multiple settings of model selection for self-supervised anomaly detection (SSAD). Let us elaborate on three self-tuning settings with increasing complexity, based on the number of anomaly types in the test data and the number of augmentation functions to search.

**1) One anomaly type + One augmentation function with unknown HPs:** Given an augmentation function, can we tune its continuous hyperparameters (HPs) in an end-to-end manner to match the (unknown) anomaly type? This is the basic setup of unsupervised model selection for SSAD, but it has not been addressed before in the literature. Our work is the first proof of concept to show that this can be done, with the help of our proposed (i) unsupervised validation loss and (ii) differentiable modeling of augmentation functions.

**2) One anomaly type + Multiple augmentation functions with unknown HPs:** Given a *set* of augmentations, can we choose the one and tune its continuous HPs to match the anomaly type? Choosing the right augmentation function from a set is a *discrete* HP search problem and can be tackled with black-box, gradient-free optimization techniques. Simplest form of this is a grid search over augmentation functions, paired with our framework to tune corresponding continuous HPs, where the model, i.e., the pair of an augmentation function and its tuned HPs, with the lowest validation loss is to be selected.

**3) Mixture of anomaly types + Multiple aug. functions with unknown HPs:** Given a *set* of $M$ augmentation functions, can we choose a subset of them and tune their continuous HPs to match the multiple anomaly types in test data? This is the hardest setting. Specifically, the fraction of samples to be drawn from each augmentation could also be set up as a HP. Then, the search in this $M$ dimensional space can be performed with approaches for black-box HP search like Bayesian Optimization (e.g. SMBO), with the goal of tuning the augmentation probabilities that yield an overall low alignment loss.

Unsupervised model selection for SSAD is a hard problem that has not been addressed by any prior work before. In this paper, we took an initial step toward addressing Setting 1 above, where Setting 2 would be a simple extension through grid search with our proposed loss. Then, Setting 3 would constitute a more generalized future work, with HP optimization for a mixture of functions, that takes our proposed framework as a subroutine. We hope to continue this line of work to make our solution closer to several real-world scenarios.

---

> ### Author Response · Authors · 2023-11-19
> **Common Responses (continued)**
>
> ## C3. Additional experiments
>
> In light of reviews, we performed 3 additional experiments: #1) varying augmentation data sizes, #2) validation-test split, and #3) contaminated training data. The results are also given in Appendix E of our revised manuscript.
>
> All experiments were done in the MVTec dataset, specifically the Cable object and the 8 anomaly types associated with it, since we didn’t have time to run experiments for all cases. The numbers in parentheses represent the ranks, and we aggregate the results over different tasks by the average accuracy and the average rank.
>
> **#1) Varying augmentation data sizes.** Our validation loss is carefully designed to be robust to $|D_{aug}| / |D_{trn}|$, since the ratio of true anomalies in test data is unknown at training time. Consequently, we simply set $|D_{aug}| = |D_{trn}|$ in all experiments, such that we perform one augmentation per training example. Specifically, instead of using all training data in every computation of the validation loss, we randomly sample 256 training samples (and 256 augmentation samples) at each computation for efficiency. The number 256 is chosen large enough to estimate the distribution of all training data.
>
> In this additional experiment, we vary the number of augmentation samples that are used in the computation of the validation loss following { 64, 128, 256, 512 }, where 256 is the choice in the original experiments. The performance is not sensitive to the size of augmented data, and we obtain similar results across eight different tasks.
>
> obj_type | ano_type | 64 | 128 | 256 (original) | 512
> --- | --- | --- | --- | --- | ---
> cable | bent_wire | 0.521 (3) | 0.596 (1) | 0.490 (4) | 0.575 (2)
> cable | cable_swap | 0.573 (2) | 0.579 (1) | 0.532 (4) | 0.573 (2)
> cable | combined | 0.913 (2) | 0.878 (4) | 0.925 (1) | 0.887 (3)
> cable | cut_inner_insulation | 0.717 (1) | 0.631 (4) | 0.667 (2) | 0.641 (3)
> cable | cut_outer_insulation | 0.457 (4) | 0.525 (1) | 0.516 (3) | 0.525 (1)
> cable | missing_cable | 0.909 (4) | 0.973 (2) | 0.998 (1) | 0.967 (3)
> cable | missing_wire | 0.507 (4) | 0.706 (3) | 0.863 (1) | 0.723 (2)
> cable | poke_insulation | 0.722 (1) | 0.683 (2) | 0.630 (4) | 0.631 (3)
> average | accuracy | 0.665 | 0.696 | 0.703 | 0.690
> average | rank | 2.6 | 2.3 | 2.5 | 2.4
>
> **#2) Validation-test split.** Although we focus on transductive learning, where unlabeled test data are given at training time, we perform an additional experiment where we split the original test data into two disjoint sets of validation and (new) test data, with the size ratio 1:1, and use only the validation data in the computation of our alignment loss. This is to create an evaluation setting where the evaluation data (i.e., new test data) are completely separated from the data observed in training time.
>
> The following table shows the result. We find a negligible difference between the two settings, demonstrating that our approach succeeds with the separated test data.
>
> obj_type | ano_type | Original | Separated
> --- | --- | --- | ---
> cable | bent_wire | 0.490 (2) | 0.577 (1)
> cable | cable_swap | 0.532 (2) | 0.648 (1)
> cable | combined | 0.925 (1) | 0.879 (2)
> cable | cut_inner_insulation | 0.667 (2) | 0.723 (1)
> cable | cut_outer_insulation | 0.516 (2) | 0.639 (1)
> cable | missing_cable | 0.998 (1) | 0.946 (2)
> cable | missing_wire | 0.863 (1) | 0.736 (2)
> cable | poke_insulation | 0.630 (2) | 0.681 (1)
> average | accuracy | 0.703 | 0.729
> average | rank | 1.6 | 1.4
>
> **#3) Contaminated training data.** We assumed a problem setting where training data consist of only normal samples, as done in many previous works on anomaly detection. However, one may consider this setting unrealistic, since acquiring pure clean data is often hard in real-world scenarios. Thus, we conduct experiments with contaminated training data, which contain a small fraction of anomalies in it. We increase the ratio of contamination up to 2\%, since the number of anomalies in the MVTec dataset is not large enough to increase more.
>
> The following table shows the result. The overall performance is similar in the three settings, showing that our approach is robust to the small fraction of anomalies in training data.
>
> obj_type | ano_type | 0% (original) | 1% | 2%
> --- | --- | --- | --- | ---
> cable | bent_wire | 0.490 (3) | 0.711 (1) | 0.581 (2)
> cable | cable_swap | 0.532 (3) | 0.642 (1) | 0.626 (2)
> cable | combined | 0.925 (1) | 0.880 (2) | 0.868 (3)
> cable | cut_inner_insulation | 0.667 (2) | 0.753 (1) | 0.585 (3)
> cable | cut_outer_insulation | 0.516 (3) | 0.603 (2) | 0.607 (1)
> cable | missing_cable | 0.998 (1) | 0.905 (2) | 0.789 (3)
> cable | missing_wire | 0.863 (1) | 0.686 (3) | 0.826 (2)
> cable | poke_insulation | 0.630 (3) | 0.649 (2) | 0.674 (1)
> average | accuracy | 0.703 | 0.729 | 0.695
> average | rank | 2.1 | 1.8 | 2.1

---

### Author Response · Authors · 2023-11-22
**Thank You for the Reviews**

Dear Reviewers,

We express our gratitude for your valuable time in reviewing our work. We are available to address any additional questions you may have regarding our work or rebuttal. Kindly inform us if there are any such concerns.

Sincerely,
Authors

---

### Meta-Review · Area_Chair_2LbE · 2023-12-05

**Metareview:**

The paper proposes, ST-SSAD, a novel approach for self-supervised anomaly detection (SSAD) that tackles the challenge of selecting appropriate data augmentation functions. It offers two main contributions: an unsupervised validation loss for tuning augmentation and differentiable augmentation functions for end-to-end hyperparameter tuning.

Points about experimental setting need to be addressed in a more convincing way, as explained below in detail. Currently, the real-world applicability of the proposed method is of concern.

**Justification For Why Not Higher Score:**

- Concerns on the setting: Relies on the entire test dataset during training and tuning, potentially limiting generalizability. Also heavily depends on prior knowledge about anomalies. The new experiments address the concerns to a degree.
- Lack of comprehensive experimental validation on the proposed components.
- Evaluates each anomaly type separately, making it unclear how to handle multiple types.
- Writing, terminology and notation issues.
- Significant improvements are not consistent.

**Justification For Why Not Lower Score:**

- Proposes an effective method for end-to-end augmentation tuning in SSAD.
- Learnable augmentations (e.g., CutDiff) have potential beyond anomaly detection.
- Well-organized and clear writing.
- Generally outperforms baselines in experiments.
- It is shown that the proposed pptimal augmentation function aligns better with real anomalies.
- End-to-end parameter optimization.
- Well-motivated and clearly presented.
- Provides theoretical support and illustrations

---

### Decision · Program_Chairs · 2024-01-16

Reject